# The efficacy of Personalized Normative Feedback interventions across addictions: A systematic review and meta-analysis

Jenny Saxton[1‡], Simone N. Rodda [1,2‡]*, Natalia Booth[1], Stephanie S. Merkouris[2], Nicki A. Dowling[2,3]

1 School of Population Health, University of Auckland, Auckland, New Zealand, 2 School of Psychology, Deakin University, Geelong, Australia, 3 Melbourne Graduate School of Education, University of Melbourne, Parkville, Australia

‡ These authors share first authorship on this work.
* s.rodda@auckland.ac.nz

**Data Availability Statement:** All relevant data are within the manuscript and its Supporting Information files.

## Abstract

Personalized Normative Feedback (PNF) may help address addictive disorders. PNF highlights discrepancies between perceived and actual peer norms, juxtaposed against self-reported behavior. PNF can be self-directed and cost-efficient. Our study estimates the efficacy of PNF alone, and in combination with other self-directed interventions, to address frequency and symptom severity of hazardous alcohol use, problem gambling, illicit drug and tobacco use. We searched electronic databases, grey literature, and reference lists of included articles, for randomized controlled trials published in English (January 2000-August 2019). We assessed study quality using the Cochrane Risk of Bias tool. Thirty-four studies met inclusion criteria (k = 28 alcohol, k = 3 gambling, k = 3 cannabis, k = 0 tobacco). Thirty studies provided suitable data for meta-analyses. PNF alone, and with additional interventions, reduced short-term alcohol frequency and symptom severity. PNF with additional interventions reduced short-term gambling symptom severity. Effect sizes were small. PNF did not alter illicit drug use. Findings highlight the efficacy of PNF to address alcohol frequency and symptom severity. The limited number of studies suggest further research is needed to ascertain the efficacy of PNF for gambling and illicit drug use. Cost-effectiveness analyses are required to determine the scale of PNF needed to justify its use in various settings.

## Introduction

Addictive behaviors associated with alcohol, tobacco, illicit drugs and gambling can have considerable negative consequences for individuals, families and the wider society. Alcohol use is the leading global risk factor for death in men and women aged 15–49 [1]. Alcohol use also causes substantial harm, attributed to 8.9% and 2.3% of disability adjusted life years in men and women 15–49, respectively [1]. Tobacco control measures and the widespread adoption of the World Health Organization's 2003 Framework Convention on Tobacco Control have been

**Funding:** This review was funded by the Health Research Council, New Zealand (17/548). https://www.hrc.govt.nz/. SR received the award. The funders had no role in study design, data collection and analysis, decision to publish, or preparation of the manuscript.

**Competing interests:** The 3-year declaration of interest statement of this research team is as follows: SR, SM and ND have received funding from multiple sources, including government departments in New Zealand and Australia. SR, SM and ND have also received funding from the International Center for Responsible Gaming (ICRG), a charitable organization, which derives its funding through contributions from multiple stakeholder groups (with funding decisions the responsibility of a scientific advisory board). ND is the recipient of a Deakin University Faculty of Health Mid-Career Fellowship. SM is the recipient of a New South Wales Office of Responsible Gambling Postdoctoral Fellowship and has formerly been the Victorian state representative (unpaid) on the NAGS Executive Committee. None of the authors have knowingly received research funding from the gambling, tobacco, or alcohol industries or any industry-sponsored organization. This does not alter our adherence to PLOS ONE policies on sharing data and materials.

linked to impressive reductions in tobacco use [2, 3]. Yet, global tobacco use prevalence remains substantial at 25%, and tobacco use was attributed to 6.4 million deaths in 2015 alone [4]. In terms of illicit drugs, cannabis dependence is the most common substance use disorder, with an estimated 22·1 million cases globally in 2016, and 31·8 million disability adjusted life years attributed to drug use overall [5]. Gambling is a common practice worldwide, though lack of data examining its relationship to health and mortality makes it difficult to estimate population level harms through metrics such as disability adjusted life years [6]. As an alternative measure, Browne and colleagues assessed eight domains of gambling-related harm to estimate decrements in health-related quality of life amongst gamblers in New Zealand [6]. Their findings indicated health-related quality of life reductions of 0.18, 0.37 and 0.54 for low-risk, moderate-risk and problem gamblers respectively, equivalent to 2.5 times the harm caused by diabetes. Gambling problems are estimated at an average of 2.3% internationally [7].

The latest edition of the Diagnostic and Statistical Manual of Mental Disorders (DSM-5) now classifies these addictive behaviors as Substance-Related and Addictive Disorders. The update takes into account new research suggesting that gambling disorder shares features of substance use disorders in terms of brain origins, clinical and physiological manifestations, comorbidity and effective treatment options [8]. Moreover, there is considerable comorbidity between these addictive behaviors [9–13]–. Psychological interventions such as cognitive-behavioral therapies (CBT) and motivational interviewing demonstrate good outcomes for individuals reporting these addictive behaviors [14–17]. Unfortunately, only 10–20% of people with addictive behaviors access face-to-face treatments, and those that do tend to have the most severe problems [18]. People with heavy, but less severe use, rarely seek help despite experiencing associated negative consequences [19].

Screening and brief interventions (SBIs), such as the Alcohol Use Disorders Identification Test (AUDIT), provide both an opportunity for identifying people with problems who are not yet seeking help and a suite of interventions that can be delivered in primary care at low cost [5]. Other Screening and brief interventions (SBIs) delivered via the internet usually last 5–20 minutes and have been reported as effective in reducing hazardous and harmful drinking [20, 21], as well as illicit drug use [22, 23] and gambling problems [24]. A 2017 Cochrane review found internet-based tobacco use interventions were significantly more effective than non-active controls at six months [25]. Effect sizes are generally smaller than more intensive, longer treatments, or those involving a clinician [25–27], but their brevity and ease of access appear attractive to those with lower levels of symptom severity [28, 29]. So whilst Screening and Brief Interventions (SBIs) may be somewhat less effective than gold standard treatments, such as CBT for gambling [16], it may still be worth offering them due to their greater cost efficiency and wider reach.

Social norms approaches are a potentially powerful and cost-effective way to promote behavior change [30] and may be incorporated into SBIs to help address the burden of addictive behaviors [31, 32]. Social norms approaches include various distinct interventions, operationalized from different aspects of social norms theory e.g., social marketing, fear-based methods, and personalized normative feedback (PNF) [30].

PNF was first developed in the United States in response to college student drinking. At this time, multiple studies had indicated that college students over-estimated the quantitative and frequency of alcohol consumption in peer groups [33]. This bias appeared stronger for personally relevant social groups (e.g., college fraternities) compared with all students or adults in general. Early studies delivering PNF with mail out questionnaires and feedback (also delivered by mail) indicated an impact on the frequency and amount of alcohol consumed [34]. The delivery mechanism changed in the early 2000's with the emergence of computer-delivered interventions. This delivery mechanism vastly increased the capacity to deliver PNF in

real time (rather than waiting for mail feedback) and allowed PNF to be more readily tailored and presented in more sophisticated formats (e.g., using graphics).

Personalized normative feedback (PNF) interventions can be considered as a subset of personalized feedback interventions (PFIs). Personalized feedback interventions (PFIs) aim to 'increase the salience of normative and personal standards in order to promote thoughtful consideration' about one's own behavior [35]. Personalized normative feedback interventions (PNF) make use of injunctive and/or descriptive normative information to elicit behavior change. Injunctive norms refer to social judgements about a particular behavior by an individual's peer group; descriptive norms refer to the prevalence of a particular behavior amongst an individual's peer group [36]. The premise of PNF is that individuals misperceive (i.e., over or underestimate) consumption levels or judgements of their peers, which contributes to maintaining their own problematic behavior. PNF asks individuals to provide information about their own consumption then presents them with the true injunctive and/or descriptive norms for their peer group. The theory behind PNF is that when confronted with their misperception of their peer group's behavior and/or the social disapproval of their peer group, an individual will adjust their own behavior towards the newly realized norm [30].

## Review rationale

Given the DSM-5 considers hazardous alcohol use, tobacco use, illicit drug and problem gambling as Substance-Related and Addictive Disorders, which are often co-occurring and share common mechanisms [37], we seek to understand whether PNF is a candidate intervention for each of these four disorders. We have chosen to focus on self-directed PNF interventions, as this will enable us to determine the efficacy of a low-cost, low resource intervention with a potentially wide reach.

Existing systematic reviews of social norms approaches to promote behavior-change, including PNF, have treated distinct personalized feedback interventions (PFIs) as though they were the same [38–40]. This prevents an understanding of which social norms approaches work best, under what circumstances, and for which problems. PNF is also frequently implemented as part of a multicomponent intervention when examining the evidence, including additional information such as official guidelines for 'safe' levels of use, or self-help strategies. In these multicomponent interventions, the mechanisms underpinning PNF could be undermined by one of the other elements (e.g., fear-based messaging) [30]. To fully understand the utility of PNF, it is important to differentiate between 'pure' PNF (i.e., PNF alone) and PNF in combination with other approaches ('mixed PNF' interventions).

While several systematic reviews have explored the efficacy of PNF to reduce alcohol, illicit drug use and gambling, largely in college populations, no single review has considered different addictive disorders with a range of sample types, and none have included tobacco as a target behavior. Furthermore, none of these systematic reviews have isolated the efficacy of self-directed PNF (alone or in combination with other self-directed interventions) to address each of the four substance-related and addictive disorders we focus on in the current review. Most existing reviews focus on PNF or other norms-based interventions delivered in-person [38, 41–48], do not target people with problems [47, 49, 50], and/or only include college student samples or samples of young people for a single addiction type [46, 47, 49, 50], which limits the generalizability of their findings. Several reviews also include studies with active control conditions [45, 47, 49, 50], which can reduce statistical power to identify intervention effects [43], whilst others allow non-randomized controlled trial (RCT) designs [49, 50], which are at higher risk of confounding and bias than RCTs [51].

### Review aims

1. To examine the efficacy of 'pure PNF' interventions (i.e., no other intervention implemented) for hazardous alcohol use, problem gambling, illicit drug and tobacco use, relative to passive control groups, in reducing frequency of use and symptom severity.

2. To examine the efficacy of PNF plus self-directed interventions ('mixed PNF interventions') for hazardous alcohol use, problem gambling, illicit drug and tobacco use, relative to passive control groups, in reducing frequency of use and symptom severity.

3. To examine whether addictive disorder type, setting (e.g., university environment), and type of additional intervention components explain the variability in the magnitude of the PNF intervention effects.

4. To examine the extent to which methodological risk of bias characteristics influence PNF intervention effects.

## Methods

Our reporting of this systematic review is compliant with the Preferred Reporting Items for Systematic reviews and Meta-Analyses (PRISMA). We published the protocol for this review in the PROSPERO database of systematic reviews [CRD42018093549], which we updated in May 2019 (original version, August 2018) [52]. Differences between the updated PROSPERO protocol and the published review include: (1) We included tobacco use as an additional addictive disorder for review; (2) We amended one study exclusion criterion: instead of excluding studies where not all participants were exposed to normative feedback, we excluded studies where not all participants were given the opportunity to access the PNF intervention. This was due to several studies reporting incomplete intervention exposure amongst participants (e.g., not all participants downloaded and used a PNF app), despite giving all participants the opportunity to do so; (3) We conducted additional sub-group analyses to explore the influence of setting, and additional intervention components on PNF efficacy; and (4) We conducted additional sensitivity analyses to assess whether papers for which we converted medians to means affected our findings.

### Search strategy

Our systematic search included an electronic database search of English language articles in EMBASE, MEDLINE, PsycINFO and the Cochrane Library Databases from January 1st 2000 (consistent with the advent of computer-delivered PNF) to August 28th 2019. Our search strategy included a combination of keywords and wildcards. This combination was intervention type (e.g., social norms, personalized feedback) AND the addictive behaviors (e.g., gambling, alcohol, drug, tobacco) AND treatment (e.g., intervention, trial). We hand-searched the reference lists of included studies. Our grey literature search comprised: (1) searching for otherwise unpublished trial data with the search terms 'alcohol and personalized feedback', 'gambling and personalized feedback', 'drug or substance and personalized feedback' and 'smoking or tobacco and personalized feedback' in the following trial registers: US ClinicalTrials.gov, Metaregister of controlled trials, and WHO International clinical trials registry platform search portal (with the exception of tobacco use, as the portal was no longer available); and (2) using the same search terms, conducting four Google searches for reports of funded projects, where the first 100 entries were examined for each search. Search terms for each database are available in S1 Appendix.

## Study eligibility criteria

Studies were eligible for inclusion if they met the following criteria: (1) were RCTs; (2) at least one arm used a PNF component, in which the feedback needed to include reference to the participant's own alcohol, gambling, illicit or prescription drug or tobacco use, and descriptive and/or injunctive normative information about alcohol, gambling, illicit drug or tobacco use; (3) the PNF intervention was delivered to individuals (not groups); (4) the PNF intervention was self-directed, however interventions where researchers instructed participants to look at personalized normative feedback were included [53], as were study debriefs by researchers [54]; (5) study samples consisted of adults (18 years and older), or mixed groups of adults and adolescents who were 16 years or older; (6) study samples consisted of individuals with some level of problematic use of alcohol, gambling, drugs or tobacco use at baseline, as determined by a screening tool, health professional, or standard definition by researchers that attempted to identify regular moderate-heavy consumption or binge consumption; (7) the study included a passive control group (i.e., no intervention, assessment only, or generic health feedback that did not include gambling, alcohol, illicit drug use or tobacco use feedback); we consider that PNF could be primarily delivered as an SBI for users not yet engaged with other interventions, for whom absolute efficacy estimates (with passive controls) are more relevant than relative efficacy estimates (with active controls); and (8) the article was published in a peer-reviewed journal in English between January 2000 and August 2019, or was identified through grey literature searches of Google or any of the three controlled trial registers listed in the search strategy section covering the same time period; and (9) studies included at least one outcome measure of alcohol, gambling, illicit drug use or tobacco use (i.e., frequency or symptom severity).

We focused on frequency and severity outcomes as they are indices that can consistently be applied across all of the included substance and behavioral addictions. Frequency (Outcome 1) was defined as how often or how frequently participants engaged in drinking, gambling, illicit drug or tobacco use in a given reference period. This may be termed as the number of drinking/gambling/illicit drug use/smoking days, occasions or episodes. We excluded outcomes measuring quantity-type frequency variables (e.g., number of drinks per week, number of bets placed) and binge-related outcomes, as they may be measuring a different construct. Measures of subjective change in drinking were also excluded, as it was unclear whether they referred to quantity or frequency (e.g., if participants were asked if their drinking had increased, decreased, or stayed the same in the past month). Symptom severity (Outcome 2) was defined as any standardized or un-standardized measure of problem severity or harm in relation to drinking, gambling, illicit drug or tobacco use. We excluded quantity measures (e.g., number of drinks consumed, dollars spent on gambling) and blood alcohol content (BAC) as they are addiction-specific and not applicable across the range of substance and behavioral addictions our review included. We also excluded measures of attitudinal change because our aim was to assess the effect of PNF on behavior change.

Articles were excluded from the current review if the: (1) PNF intervention primarily targeted weight loss, or any other health behaviors that did not include alcohol, gambling, illicit drug or tobacco use; (2) intervention was labelled as PNF, but descriptive or injunctive normative feedback was not provided; (3) PNF intervention targeted people with specific physical or psychological comorbidities (e.g., war veterans with post-traumatic stress disorder), which are less generalizable to other populations; (4) PNF intervention was delivered in a group setting; (5) PNF intervention was a prevention program designed only for people not yet engaged in problem alcohol consumption, gambling, illicit drug or tobacco use (sometimes referred to as 'at risk' groups) or if subgroup analyses based on severity level were conducted, where groups

were identified after randomization [55]; (6) participants received any formal treatment (psychological or drug therapy) in conjunction with PNF, or who were within 12 months of completing formal treatment for problem use/addictive disorder; (7) participants were mandated to complete the program (e.g., for legal reasons); (8) sample comprised only children or adolescents younger than 16 years; (9) studies did not assess relevant outcome measures (i.e., alcohol, gambling, drug or tobacco frequency or severity); (9) studies provided insufficient information about the intervention for it to be categorized or provided no usable data; (10) article was a review, conference proceeding, abstract, book or book chapter, or protocol; (11) studies included active comparison/control groups (e.g., relevant health information given as leaflet, or where participants rated the usefulness of relevant self-help information); (12) studies required in-person contact with a researcher, facilitator or health professional for all intervention arms, or where participants were given verbal feedback about their scores; we deliberately excluded interventions requiring in-person contact as our interest was focused on estimating the efficacy of PNF as a very low cost, low resource intensive intervention; and (13) participants in the intervention arm were not all given the opportunity to take part in the PNF intervention, or where it was not possible to verify that the majority of participants were given access to the intervention.

## Article screening, data extraction, and quality assessment

Two reviewers independently screened all titles and abstracts of articles retrieved by the literature searches against the inclusion and exclusion criteria. Discrepancies between reviewers were resolved through discussion or by a third reviewer if an agreement could not be reached. Full text review was also conducted by two reviewers independently. Discrepancies were again resolved by a third reviewer if necessary.

Data were independently extracted by two reviewers using a standardized extraction sheet in Microsoft Excel. The data extracted included comprehensive details about the study characteristics (e.g., year of publication, sample type), full descriptions of the intervention and control groups and outcome measures (e.g., type of measure employed, means and standard deviations). Any variations in the data extracted by the two reviewers were resolved through discussion, involving a third reviewer when necessary. Studies with missing data were not requested from study authors.

We used the Cochrane Risk of Bias tool, version 2.0 [56] to assess studies for risk of bias. The tool assesses papers for potential bias on five domains: (1) the randomization process; (2) bias due to deviations from the intended interventions; (3) bias due to missing outcome data; (4) bias in measurement of the outcome; and (5) bias in selection of the reported result. Each domain is scored as low, some concerns or a high risk of bias. If there are some concerns for one domain only, the overall paper is judged to have some concerns; if the paper has some concerns in two or more domains, or high risk in one or more domain, the overall judgement is the paper is at high risk of bias.

Two reviewers independently assessed the same one-third of the papers to ensure consistent application of the tool. After resolving any discrepancies, the remaining two-thirds of papers were assessed by one reviewer. All papers judged to have some concerns or a high risk of bias were also double-assessed. We used original articles and their published protocols (where available) to arrive at our Risk of Bias judgements.

## Data synthesis

**Description of included studies and meta-analysis.** We briefly described the characteristics and tabulated the results of all included studies, considering pure and mixed PNF studies

separately. We then ran a series of meta-analyses of studies providing suitable data, which were performed in Review Manager (version 5.3), with forest plots created in STATA (version 13). Our main analyses consisted of random-effects models with the inverse variance method to generate standardized mean differences (SMD), with 95% confidence intervals, based on follow up means and standard deviations (SDs) for each of the continuous outcomes (frequency and symptom severity), four follow up periods, and for the pure and mixed PNF studies separately. Conventional thresholds were used to label effect sizes as small (0.2), medium (0.5) or large (0.8) [57]. Heterogeneity among studies was estimated using Chi square and associated P-value, and the $I^2$ statistic. Adapting guidance from the Cochrane Review Handbook, we considered heterogeneity to be minimal if $I^2$ was 0–40% and the Chi square p-value was not significant (p>0.1), moderate if $I^2$ was 41–60% with a significant Chi square p-value (p≤0.10), and substantial if $I^2$ was 61–100% with a significant Chi square p-value [58]. A minimum of two estimates were required to conduct a meta-analysis.

Our full list of decision rules for meta-analytic estimates and statistical conversions is provided in S2 Appendix, but the key rules are summarized as follows: (1) If more than one frequency or severity outcome was reported in the same article, preference was given to measures employed more frequently, followed by complete tools rather than sub-scales, then standardized over unstandardized tools, and multi-dimensional over single dimensional tools (for symptom severity) or 'days' followed by 'occasions', followed by 'episodes' (for frequency); (2) If no means or SDs were available, we calculated them as per the conversion formulae available in the Cochrane handbook, when possible [59]. Using the same guidance material, we calculated single intervention means, SDs and numbers of participants where multiple intervention means, SDs and numbers of participants were reported. Where possible, we also converted medians to means according to Hozo's guidance [60] but given that conversions of medians to means is not yet standard practice for systematic reviews, we conducted sensitivity analyses omitting these studies to assess any difference in results. We also obtained estimates of relevant data from published graphs if necessary. Where data could not be converted or estimated, studies were excluded from the meta-analysis and reported in the description of studies section only; (3) Intention to treat data were preferred over completer data; (4) Data from the least adjusted models were preferred over more adjusted; (5) Overall results were preferred over males and females separately; (6) If an article presented two or more values within a single follow up period (as defined in our review), we used data from the longest follow up period; (7) If multiple papers were available for the same data set, we used the article reporting our preferred outcomes, then with the longest follow up period, then with the least adjusted results, then using the most robust measurement tool.

## Subgroup analyses

Pre-specified subgroup analyses were performed to investigate potential differences in PNF intervention effects according to the following study characteristics: (1) addiction type (alcohol, gambling, illicit drugs, tobacco use); (2) sample type (university/college students; non-university/college students); and (3) type of additional intervention component included, using common categories emerging from the articles (mixed PNF studies only).

## Sensitivity analyses

To examine the influence of methodological characteristics on the PNF intervention effects, we excluded articles rated overall as having 'some concerns' or 'high risk of bias' from the main analyses. We also sought to examine whether excluding papers for which we had converted medians to means changed the findings from our main analyses.

## Results

### Search results

5,171 articles were screened from primary and secondary (reference list) searches after the removal of duplicates, with 232 full text articles reviewed. Of these, 34 studies were included in the final review, with 30 of these studies providing sufficient data for inclusion in the meta-analyses. Full details of the literature search results are presented in the PRISMA diagram in Fig 1.

### Characteristics of included studies

Pure and mixed PNF study characteristics, study results and risk of bias assessments are presented in Tables 1 and 2 respectively. Further details of specific intervention elements and the number of studies using them are provided in S3 Appendix.

### Number of studies and focus behaviors

Thirteen studies examined the efficacy of at least one pure PNF arm against a passive control. Twelve of these focused on alcohol, and one on gambling. We did not identify any pure PNF studies of illicit drug or tobacco use that met our inclusion criteria.

Twenty-four studies tested the efficacy of PNF combined with other self-directed intervention components against a passive control. Three of these also included pure PNF arms [65, 70, 90]. Of these, 19 studies focused on alcohol, two on gambling, and three on illicit drug use, specifically cannabis. We did not identify any mixed PNF studies of tobacco use that met our inclusion criteria.

### Risk of bias

The majority of pure PNF studies were judged to be at low risk of bias (k = 10, 76.9%), with three exceptions, which were all found to have 'some concerns' [32, 63, 70]. Most mixed PNF studies were judged to be at low risk of bias (k = 18, 75%), with five studies judged to have 'some concerns' overall [53, 70, 73, 85, 88] and one study judged to be at overall high risk of bias [87]. Domain 1 –risk of bias about the randomization process—was the most common area where papers were considered to have some risk of bias, and this was generally due to lack of detail rather than authors actively stating inappropriate group allocation.

### Meta-analyses

The results of the main meta-analyses are as forest plots presented in Figs 2–5. These figures include the SMDs and 95% CIs for individual studies within each analysis, as well as the pooled effects and heterogeneity statistics for each of the main analyses. The results of all subgroup analyses related to addiction type, setting and additional intervention components (mixed PNF studies) and the sensitivity analyses are presented in S4 Appendix for pure PNF studies, and S5 Appendix for mixed PNF studies.

### Pure PNF vs control

**Frequency: Main analyses.** As shown in Fig 2, there were no significant differences between pure PNF intervention and the control groups on frequency at 0–3 months (k = 6), 4–11 months (k = 6) or 12–23 months (k = 2) post-baseline. Heterogeneity for 0–3 and 4–11 month follow up periods was substantial, and was minimal for 12–23 months. There were no studies available for the follow up period ≥24 months.

## PRISMA 2009 Flow Diagram

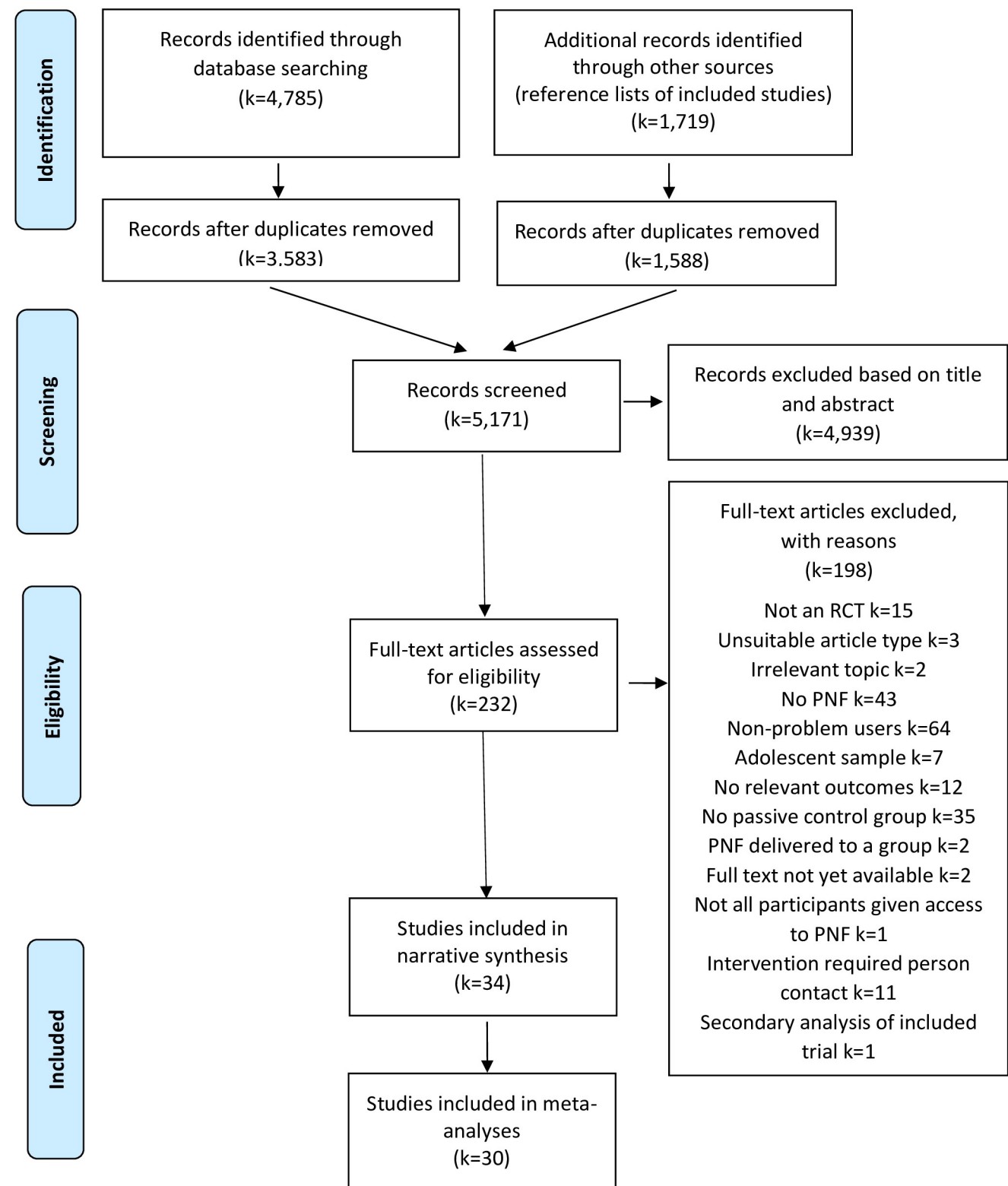

**Fig 1. PRISMA diagram for systematic review and meta-analysis.**

**Table 1. Characteristics of included pure Personalized Normative Feedback (PNF) studies (k = 13).**

| Author & date | Country | Setting | Sample size | Age | Sex | Ethnicity | Problem-related inclusion criteria | Intervention | Outcomes | Follow up | Direction of between group effects[a] | Risk of Bias[b] |
|---|---|---|---|---|---|---|---|---|---|---|---|---|
| Alcohol studies | | | | | | | | | | | | |
| Collins et al. (2014) [61] | USA | University | N = 473 | Mean 20.8 years (SD = 1.4) | 56% female | 67.1% White, 17.8% Asian, 9.6% mixed, 1% Black/ African American, Other groups 4.5% | ≥4 drinks (women) ≥5 drinks (men) on a single occasion in last 30 days | Single pure PNF arm using descriptive norms | Frequency: number of drinking days last month | 1, 6, 12 months | Frequency: +ve at 1 month, no effect at 6 months | Low |
| | | | | | | | | | Severity: RAPI[d] score | | Severity: +ve at one month, no effect at 6 months | |
| LaBrie et al. (2013) [62] [e] | USA | University | N = 1,663 | Mean 19.9 years (SD = 1.3) | 56.7% female | 75.7% White | ≥4 drinks (women) ≥5 drinks (men) on a single occasion in last month | 8 pure PNF arms with increasingly specific reference groups using descriptive norms (treated as one in analyses) | Frequency: number of drinking days last month | 1, 3, 6, 12 months | Frequency: +ve over 12 months | Low |
| | | | | | | | | | Severity: RAPI score | | Severity: +ve over 12 months | |
| Lewis & Neighbors (2007) [63] | USA | University | N = 185 | Mean 20.1 years (SD = 1.8) | 54.6% female | 97.3% White, 2.7% other groups | ≥4 drinks (women) ≥5 drinks (men) on a single occasion in last month | 2 pure PNF arms with either gender-specific or gender neutral feedback, using descriptive norms | Frequency: average number of drinking days per week last month | 1 month | Frequency: +ve for both groups | Some concerns |
| | | | | | | | | | Severity: mean Alcohol Consumption Inventory score | | Severity: +ve for both groups | |
| Lewis et al. (2007) [64] | USA | University | N = 316 | Mean 18.5 years (SD = 2.0) | 52% female | 99.6% White | ≥4 drinks (women) ≥5 drinks (men) on a single occasion in last month | 2 pure PNF arms with either gender-specific or gender neutral feedback, using descriptive norms for freshmen | Frequency: Number of drinking days per typical week | 5 months | Frequency: +ve for both groups | Low |
| Lewis et al. (2014) [65] [e] | USA | University | N = 240 | Mean 20.1 years (SD = 1.5) | 57.6% female | 70% White, 12.5% Asian, 16.2% other or not indicated | ≥4 drinks (women) ≥5 drinks (men) on a single occasion in last month | Single pure PNF arm using descriptive norms | Frequency: average number of drinking days last month | 3 and 6 months | Frequency: +ve at both time points | Low |
| | | | | | | | | | Severity: BYAACQ[f] problem score | | Severity: no effect at both time points | |
| Miller et al. (2018) [32] | USA | Young adult veterans, remote access | N = 784 | Mean 28.9 years (SD = 3.3) | 17% female | 84% White | AUDIT[g] score of ≥3 (women) or ≥4 (men) | Single pure PNF arm using descriptive norms | Severity: modified BYAACQ score (black out item removed) | 1 month | Severity: +ve | Some concerns |

(*Continued*)

**Table 1.** (Continued)

| Author & date | Country | Setting | Sample size | Age | Sex | Ethnicity | Problem-related inclusion criteria | Intervention | Outcomes | Follow up | Direction of between group effects[a] | Risk of Bias[b] |
|---|---|---|---|---|---|---|---|---|---|---|---|---|
| Neighbors et al. (2004) [66] | USA | University | N = 252 | Mean 18.5 years (SD = 1.2) | 58.7% female | 79.5% White, 13.7% Asian/ Asian American, 6.8% other groups | ≥4 drinks (women) ≥5 drinks (men) on a single occasion in last month | Single pure PNF arm using descriptive norms | Severity: composite score: Alcohol Consumption Index, RAPI score, mean drinks per week, highest number of drinks last month on a single occasion | 3 and 6 months | Severity: +ve at both time points | Low |
| Neighbors et al. (2006) [67] | USA | University | N = 214 | Mean 19.7 years (SD = 2.0) | 56% female | 98% White, 2% other groups | ≥4 drinks (women) ≥5 drinks (men) on a single occasion in last month | Single pure PNF arm using descriptive norms | Severity: modified RAPI | 2 months | Severity: no effect | Low |
| Neighbors et al. (2010) [68] | USA | University | N = 818 | Mean 18.16 years (SD = 0.6) | 57.6% female | 65.3% White, 24.2% Asian/ Pacific Islander, 4.2% Hispanic/ Latino, 1.5% African American, remainder Other groups | ≥4 drinks (women) ≥5 drinks (men) on a single occasion in last month | Four pure PNF arms using descriptive norms: 1) gender-specific PNF once and (2) biannually; (3) gender-nonspecific PNF once and (4) biannually; | Severity: modified RAPI score | 6, 12, 18, 24 months[h] | Severity: no effect at 6 months. Women in gender-specific feedback groups +ve over time, no effect in men. Non-gender-specific feedback group = -ve over time | Low |
| Neighbors et al. (2016) [54] | USA | University | N = 992 | Mean 20.6 years (SD = 1.7) | 53% female | 62% White, 16% Asian, 5% Black, 8% Mixed 1% Native American, 1% Native Hawaiian/ Pacific Islander, 7% Other | ≥4 drinks (women) ≥5 drinks (men) on a single occasion in last month | Two pure PNF arms: 1) explicit correction of descriptive drinking norms or 2) no explicit correction of misperceived descriptive drinking norms | Frequency: number of drinking days last month | 3 and 6 months | Frequency both groups: +ve at 3 months, no effect at 6 months | Low |
| | | | | | | | | | Severity: YAAPST[i] problem score | | Severity: no effect at 3 or 6 months | |

(*Continued*)

**Table 1.** (Continued)

| Author & date | Country | Setting | Sample size | Age | Sex | Ethnicity | Problem-related inclusion criteria | Intervention | Outcomes | Follow up | Direction of between group effects[a] | Risk of Bias[b] |
|---|---|---|---|---|---|---|---|---|---|---|---|---|
| Neighbors et al. (2018) [69] | USA | University | N = 959 | Mean 21.47 years (SD = 2.0) | 54% female | 27% White, 24% Asian, 18% Black/African American, 31% Hispanic | ≥4 drinks (women) ≥5 drinks (men) on a single occasion in last month | Eight pure PNF arms using a mixture of injunctive and descriptive norms; Content varied by type of message framing (common or uncommon; healthy or unhealthy; positively or negatively viewed by others) | Severity: modified RAPI score | 3 and 6 months | Severity: no overall PNF or subgroup effects | Low |
| Young & Neighbors (2019) [70][e] | USA | University | N = 250 | Mean 21.02 years (SD = 2.2) | 70.4% | 44.5% White 1.6% Native American/ American Indian, 12.1% Black/African American, 22.7% Asian, 1.2% Native Hawaiian/ Pacific Islander, 4.9% Multiethnic, 13.0% Other | ≥4 drinks (women) ≥5 drinks (men) on a single occasion in last month | Single pure PNF arm using descriptive norms | Severity: RAPI and BYAACQ scores | 1 month | Severity: no effect of pure PNF on latent variable combining RAPI and BYAACQ scores | Some concerns |
| Gambling studies | | | | | | | | | | | | |
| Neighbors et al. (2015) [71] | USA | University | N = 252 | Mean 23.11 years (SD = 5.3) | 40.5% female | 33.4% White, 39.4% Asian, 10.8% African American, 5.2% mixed, 11.2% Other groups | ≥2 on SOGS[j] scale | Single pure PNF arm using descriptive norms | Frequency: number of days gambled last 12 months; Severity: Gambling Problems Index score | 3 and 6 months | Frequency: no effect at 3 or 6 months; Severity: +ve at 3 months, no effect at 6 months | Low |

a Direction of between group effects: No effect = p>0.05; +ve = beneficial effect of intervention on outcome compared to control; -ve = worsening of outcome in intervention group compared to control

b Risk of Bias: RoB 2: A revised Cochrane risk-of-bias tool for randomized trials

c SD = Standard Deviation

d RAPI: Rutgers Alcohol Problem Index (White & Labouvie, 1989)

e Study also included at least one mixed PNF arm, and is included in Table 2

f BYAACQ: Brief Young Adult Alcohol Consequences Questionnaire (Kahler, Strong, & Read, 2005)

g AUDIT: Alcohol Use Disorders Identification Test (World Health Organization, 2001); AUDIT = full 10 item measure; AUDIT-C = 3 consumption items

h Means and SDs were not extractable from the paper due to poor resolution of graphs

I YAAPST: Young Adult Alcohol Problems Screening Test (Hurlbut & Sher, 1992)

j SOGS: South Oaks Gambling Screen (Lesieur & Blume, 1987).

**Table 2. Characteristics of included mixed Personalized Normative Feedback (PNF) studies (k = 24).**

| Author & date | Country | Setting | Sample size | Age | Sex | Ethnicity | Problem-related inclusion criteria | Intervention | Outcomes | Follow up | Direction of between group effects[a] | Risk of Bias[b] |
|---|---|---|---|---|---|---|---|---|---|---|---|---|
| **Alcohol studies** | | | | | | | | | | | | |
| Andersson et al. (2015) [72] | Sweden | University | N = 1,678 | Mean 23.2 years (SD[c] = 2.9) | 41% female | Not reported | AUDIT[d] score: hazardous drinking | 4 PNF+ arms using descriptive norms. Identical content, but varied in mode of delivery: Web-based (written only) and Interactive Voice Response (automated audio). Both modes were delivered once or repeated | Frequency: number of drinking days per week | 6 weeks | Frequency: No effect of any group in intention to treat analyses. | Low |
| | | | | | | | | Additional content: Presentation of negative consequences, Tips and tools for cutting down | Severity: AUDIT score | | Severity: +ve for all PNF groups | |
| Baldin et al. (2018) [73] | Brazil | Nightclubs | N = 465 | Mean 24.7 years (SD = 6.0) | 35.5% female | Not reported | AUDIT score ≥ 8 | Single mixed PNF arm using descriptive norms | Severity: lack of control over drinking (binary variable) | 6 months | No effect | Some concerns |
| | | | | | | | | Additional content: Presentation of negative consequences, financial and time costs associated with the behavior, tips and tools for cutting down | | | | |
| Bendtsen et al. (2015) [74] | Sweden | University health centers | N = 1605 | 70% were 18–20 years | Approx. 50% female | Not reported | Heavy episodic drinking > once per month or >14 standard drinks (men) or 9 (women) per week | Single mixed PNF arm using descriptive norms | Frequency: number of drinking days per week | 2 months | No effect | Low |
| | | | | | | | | Additional content: Presentation of negative consequences, tips and tools for cutting down, information provision | | | | |
| Bertholet et al. (2015) [75] | Switzerland | Army recruitment centers | N = 737 | Mean 20.75 Years (SD = 1.1) | 100% male | Not reported | >14 drinks/week or ≥6 drinks/occasion ≥monthly or AUDIT score ≥8 | Single mixed PNF arm using descriptive norms | Severity: AUDIT score and number of alcohol consequences[e] | 6 months | +ve effect on AUDIT score, no effect on number of alcohol consequences | Low |
| | | | | | | | | Additional content: Presentation of negative consequences, information provision | | | | |
| Bertholet et al. (2019) [26] | USA | Online, Amazon Mechanical Turk survey respondents | N = 977 | Mean 34.2 years (SD = 9.8) | 45% female | 80.4% White | AUDIT score ≥8 and ≥15 drinks per week | Single mixed PNF arm using descriptive norms | Severity: Sum of 11 possible negative consequences of alcohol use (non-standardized measure) | 6 months | No effect | Low |
| | | | | | | | | Additional content: Presentation of negative consequences, tips and tools for cutting down, information provision | | | | |
| Butler & Correia (2009) [53] | USA | University | N = 56 | Mean 20 years | 63–65% female | 86–96% White | ≥2 episodes drinking ≥5 drinks (males) or ≥4 (females), and 2 alcohol related problems in the past 28 days | Single mixed PNF arm using descriptive norms | Frequency: number of drinking occasions/last 28 days | 4 weeks | Frequency: +ve | Some concerns |
| | | | | | | | | Additional content: Presentation of negative consequences, financial and time costs associated with the behavior, tips and tools for cutting down, information provision | Severity: RAPI[f] score | | Severity: no effect | |

(Continued)

**Table 2.** (Continued)

| Author & date | Country | Setting | Sample size | Age | Sex | Ethnicity | Problem-related inclusion criteria | Intervention | Outcomes | Follow up | Direction of between group effects[a] | Risk of Bias[b] |
|---|---|---|---|---|---|---|---|---|---|---|---|---|
| Cunningham et al. (2012) [77] | Canada | University | N = 425 | Mean 22.6 years (SD = 3.9) | 47.5% female | Not reported | AUDIT-C score ≥4 | Single mixed PNF arm using descriptive norms / Additional content: Presentation of negative consequences, Tips and tools for cutting down, exploration of participant's current feelings and opinions about their behavior | Severity: AUDIT-C score | 6 weeks | No effect | Low |
| Johnson et al. (2018) [78] | Australia | Outpatients waiting room | N = 837 | Mean 44.0 years (SD = 17.4) | 25% female | Not reported | AUDIT-C score of 5–9 | Single mixed PNF arm using descriptive norms / Additional content: Presentation of negative consequences, financial costs associated with the behavior, tips and tools for cutting down, information provision | Frequency: number of drinking days last week; Severity: AUDIT score | 6 and 12 months | Frequency: no effect at either time point; Severity: no effect at either time point | Low |
| Kypri et al. (2009) [79] | Australia | University | N = 2435 | Mean 19.7 years (SD = 1.8) | 45% female | Not reported | AUDIT score ≥8, and consumed ≥4 standard drinks (women) or ≥6 (men) on a single occasion in last 4 weeks | Single mixed PNF arm using descriptive norms. Booster at 1 month follow up, comparing current drinking levels with baseline / Additional content: Presentation of negative consequences, financial costs associated with the behavior, tips and tools for cutting down, information provision | Frequency: number of drinking days last month; Severity: Alcohol problems score and AREAS[g] score | 1 and 6 months | Frequency: +ve at both time points; Severity: no effect for either measure at either time points | Low |
| Kypri et al. (2013) [80] | New Zealand | University | N = 1789 | Mean 20.1 years (SD = 1.7) | Approx. 70% female | 100% Maori | AUDIT-C score ≥4 | Single mixed PNF arm using descriptive norms / Additional content: Presentation of negative consequences, financial costs associated with the behavior, tips and tools for cutting down, information provision | Frequency: number of drinking days last month; Severity: AREAS score | 5 months | Frequency: +ve; Severity: +ve | Low |
| Kypri et al. (2014) [81] | New Zealand | University | N = 3422 | Mean 20 years (SD = 1.8) | 58% female | 0% Maori (no other details) | AUDIT-C score ≥4 | Single mixed PNF arm using descriptive norms / Additional content: Presentation of negative consequences, financial costs associated with the behavior, tips and tools for cutting down, information provision | Frequency: number of drinking days last month; Severity: AREAS score | 5 months | Frequency: no effect; Severity: no effect | Low |

(Continued)

**Table 2.** (Continued)

| Author & date | Country | Setting | Sample size | Age | Sex | Ethnicity | Problem-related inclusion criteria | Intervention | Outcomes | Follow up | Direction of between group effects[a] | Risk of Bias[b] |
|---|---|---|---|---|---|---|---|---|---|---|---|---|
| Murphy et al. (2015) [82] | USA | University | N = 87 | Mean 18.6 years (SD = 1.2) | 48.6% female | 64.3% White, 29.5% African American, remaining % Other groups | ≥1 heavy drinking episode last month (5 drinks for men, 4 for women on a single occasion) | Single mixed PNF arm using descriptive norms. Additional content: Presentation of negative consequences, financial costs associated with the behavior | Severity: Number of alcohol related consequences | 1 and 6 months | No effect at either time point | Low |
| Ridout & Campbell (2014) [83] | Australia | University | N = 98 | Mean 19.05 years (SD = 1.8) | 78% female | 56% White | AUDIT score ≥8 | Single mixed PNF arm using descriptive norms. Booster at 1 month including participant's percentile rank for drinking behaviors and how their consumption had changed since baseline. Additional content: Presentation of negative consequences, tips and tools for cutting down. Booster at 1 month informing participant of associated health impacts of any change observed since baseline | Frequency: number of drinking days last month | 1 and 3 months | +ve at both time points | Low |
| Wagener et al. (2012) [84] | USA | University | N = 152 | Mean 20.9 years (SD = 1.9) | 45.4% female | 84.6% White | ≥1 heavy drinking episode last month (5 drinks men, 4 women on a single occasion), usually consumes ≥20 drinks/month, and reported ≥1 associated negative consequence last month | Single mixed PNF arm using descriptive norms. Additional content: Presentation of negative consequences, financial costs associated with the behavior, exploration of participant's current feelings and opinions about their behavior | Severity: B-YAACQ[h] score | 10 weeks | No effect | Low |
| Walters et al. (2000) [85] | USA | University | N = 28 | 19.7 years (SD = 1.5) | 40% female | 62% White, 30% Hispanic, 8% Other groups | Consumed >40 drinks the previous month | Single mixed PNF arm using descriptive norms. Additional content: Presentation of negative consequences, financial costs associated with the behavior | Severity: Short index of Problems score | 6 weeks | No effect | Some concerns |
| Walters et al. (2009) [86] | USA | University | N = 136 | Mean 19.8 years | 64.2% female | 84.6% White | ≥1 heavy drinking episode in last 2 weeks (5 drinks for men, 4 for women on a single occasion) | Single mixed PNF arm using descriptive norms. Additional content: Presentation of negative consequences, financial costs associated with the behavior, information provision | Severity: RAPI score and composite severity measure created by authors | 3 and 6 months | No effect of either measure at either time point | Low |

(Continued)

**Table 2.** (Continued)

| Author & date | Country | Setting | Sample size | Age | Sex | Ethnicity | Problem- related inclusion criteria | Intervention | Outcomes | Follow up | Direction of between group effects[a] | Risk of Bias[b] |
|---|---|---|---|---|---|---|---|---|---|---|---|---|
| LaBrie et al. (2013) [62][i] | USA | University | N = 183 PNF, n = 184 control | Mean 19.9 years (SD = 1.3) | 56.7% female | 75.7% White | ≥4 drinks (women) ≥5 drinks (men) on a single occasion in last month | Single mixed PNF arm using descriptive norms | Frequency: number of drinking days last month | 1, 3, 6, 12 months | Frequency: +ve over 12 months | Low |
| | | | | | | | | Additional content: Presentation of negative consequences, financial costs associated with the behavior, tips and tools for cutting down, information provision | Severity: RAPI score | | Severity: no effect over 12 months | |
| Lewis et al. (2014) [65][i] | USA | University | N = 240 | Mean 20.1 years (SD = 1.5) | 57.6% female | 70% White, 12.5% Asian, 16.2% other or not indicated. | ≥4 drinks (women) ≥5 drinks (men) on a single occasion in last month | Single mixed PNF arm using descriptive norms | Frequency: average number of drinking days last month | 3 and 6 months | Frequency: +ve at both time points | Low |
| | | | | | | | | Additional content: Presentation of negative consequences | Severity: BYAACQ problem score | | Severity: no effect at either time point | |
| Young & Neighbors (2019) [70][i] | USA | University | N = 250 | Mean 21.02 years (SD = 2.2) | 70.4% female | 44.5% White 1.6% Native American/ American Indian, 12.1% Black/African American, 22.7% Asian, 1.2% Native Hawaiian/ Pacific Islander, 4.9% Multiethnic, 13.0% Other | ≥4 drinks (women) ≥5 drinks (men) on a single occasion in last month | Single mixed PNF arm using descriptive norms | Severity: RAPI and BYAACQ scores | 1 month | Severity: +ve on latent variable combining RAPI and BYAACQ scores | Some concerns |
| | | | | | | | | Additional content: Exploration of participant's current feelings and opinions about their behavior | | | | |
| **Gambling studies** | | | | | | | | | | | | |
| Luquiens et al. (2016) [87] | France | Online (poker players) | N = 557 | Mean 34.7 years (SD = 10.1) | 8% female | Not reported | PGSI[f] score ≥5 | Single mixed PNF arm using descriptive norms | Frequency: number of gambling sessions (last 30 days) | 6 and 12 weeks | Frequency: no effect at either time point | High |
| | | | | | | | | Additional content: Presentation of negative consequences | Severity: PGSI score | | Severity: no effect at either time point | |
| Martens et al. (2015) [88] | USA | University | N = 220 | Mean PNF 21.69 years (SD = 3.6) mean control 21.84 years (SD = 5.0) | 38% female (PNF) 41% (control) | White (80% PNF; 77% control), Asian/ Asian American (5% PNF, 10% control), African American (6% PNF, 7% control), Hispanic (5% PNF, 4% control) | Gambling ≥ once in past 60 days and either: ≥3 on the South Oaks Gambling Screen or ≥1 on the Brief Biosocial Gambling Screen. | Single mixed PNF arm using descriptive norms | Frequency: number of days gambled in the last 60 days | 3 months | Frequency: no effect | Some concerns |
| | | | | | | | | Additional content: Presentation of negative consequences, Financial and time costs associated with the behavior, Tips and tools for cutting down, Exploration of participant's current feelings and opinions about their behavior | Severity: PGSI score | | Severity: +ve | |
| **Illicit drug studies** | | | | | | | | | | | | |

(*Continued*)

**Table 2.** (Continued)

| Author & date | Country | Setting | Sample size | Age | Sex | Ethnicity | Problem-related inclusion criteria | Intervention | Outcomes | Follow up | Direction of between group effects[a] | Risk of Bias[b] |
|---|---|---|---|---|---|---|---|---|---|---|---|---|
| Elliott et al. (2014) [89] | USA | University | N = 317 | 18–23 years of age | 52% female | 78% White | Used marijuana in the last month | Two mixed PNF arms using descriptive and injunctive norms (treated as one in analyses): group 1 underwent partial pre-intervention assessments not measuring marijuana use; group 2 underwent full assessment, including measurement of marijuana use | Frequency: Number of marijuana use days in the last month | 1 month | Frequency: no effect | Low |
| | | | | | | | | Additional content: Financial costs associated with the behavior, tips and tools for cutting down, exploration of participant's current feelings and opinions about their behavior, information provision | Severity: Rutgers Marijuana Problems Inventory score, number of marijuana abuse symptoms, number of marijuana dependence symptoms (DSM–IV)[k] | | Severity: no effect for any of the three outcomes | |
| Lee et al. (2010) [90] | USA | University | N = 341 | Mean 18 years (SD = 0.3) | 55% female | 68.3% White, 15.5% Asian, 1.5% African American, 6.2% Hispanic, remaining % Other groups or not given | Use of marijuana in the 3 months before screening | Single mixed PNF arm using descriptive norms | Frequency: Number of marijuana use days in the last month | 3 and 6 months | Frequency: no effect at either time point | Low |
| | | | | | | | | Additional content: Exploration of participant's current feelings and opinions about their behavior, Tips and tools for cutting down | Severity: Rutgers Marijuana Problem Index score | | Severity: no effect at either time point | |
| Palfai et al. (2014) [91] | USA | University | N = 123 | Mean 19–20 (SD 1.1–1.3) | 54% female | 87% White, 2.4% Black, 1.6% American Indian/Alaskan, 5.7% Asian | At least monthly marijuana use in the last 90 days | Two mixed PNF arms using descriptive norms: group 1 onsite, group 2 offsite participation | Frequency: number of days used marijuana in last 30 days | 3 and 6 months | Frequency: no effect at either time point | Low |
| | | | | | | | | Additional content: Presentation of negative consequences, financial costs associated with the behavior, tips and tools for cutting down | Severity: The Marijuana Problems Scale score | | Severity: no effect at either time point | |

a Direction of between group effects: No effect = p>0.05; +ve = beneficial effect of intervention on outcome compared to control; -ve = worsening of outcome in intervention group compared to control

b Risk of Bias: RoB 2: A revised Cochrane risk-of-bias tool for randomized trials

c SD = Standard Deviation

d AUDIT: Alcohol Use Disorders Identification Test (World Health Organization, 2001); AUDIT = full 10 item measure; AUDIT-C = 3 consumption items

e Number of Alcohol Consequences (Wechsler et al. 1994)

f RAPI: Rutgers Alcohol Problem Index (White & Labouvie, 1989)

g AREAS: Academic Role Expectations and Alcohol Scale (McGee & Kypri, 2004)

h BYAACQ: Brief Young Adult Alcohol Consequences Questionnaire (Kahler, Strong, & Read, 2005)

i Study also included at least one pure PNF arm, and is included in Table 1

j PGSI: Problem Gambling Severity Index, a subscale of the Canadian Problem Gambling Index (Ferris & Wynne, 2001)

k DSM-IV: Diagnostic and statistical manual of mental disorders (4th ed., Text Revision) (American Psychiatric Association, 2000)

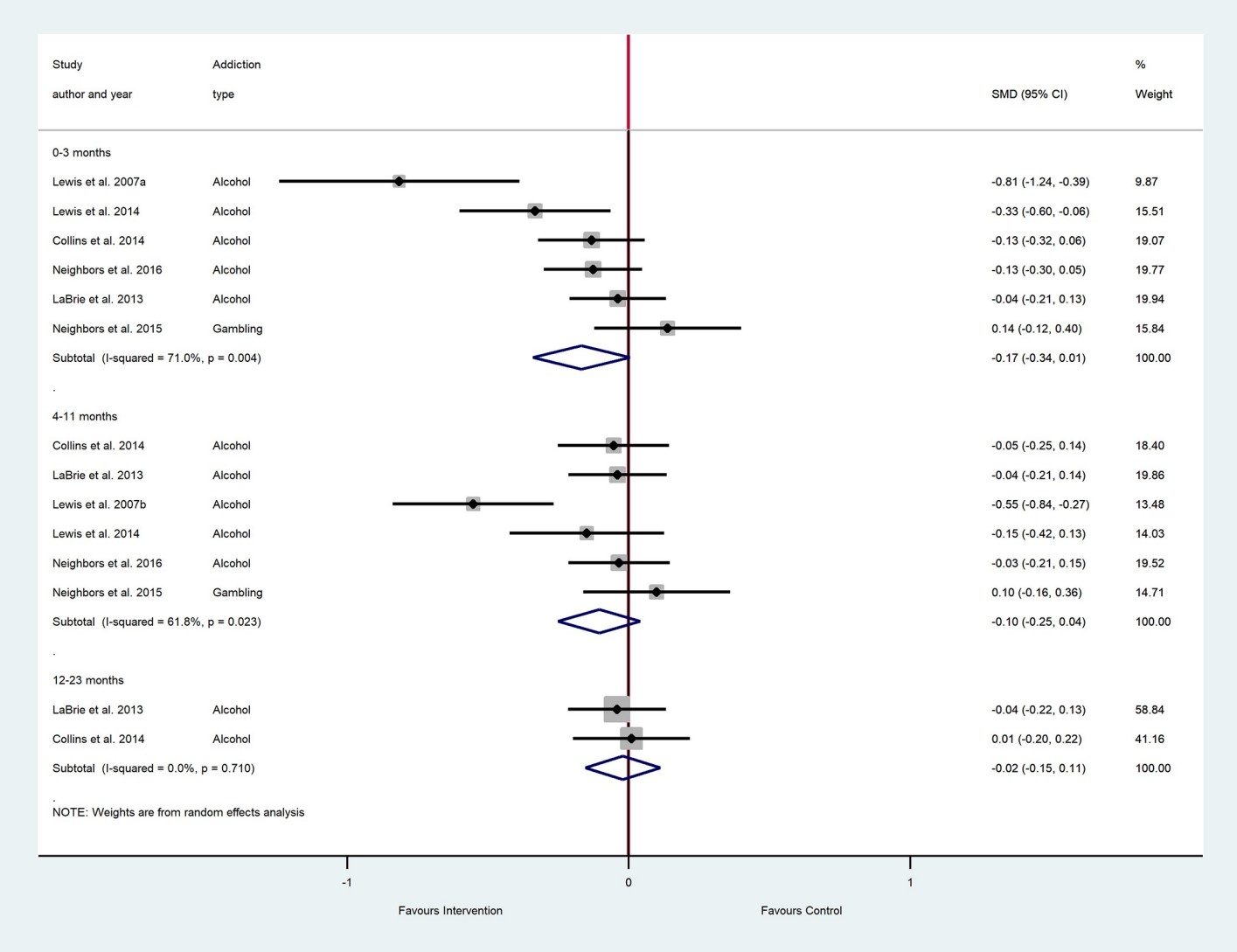

**Fig 2. The efficacy of pure PNF on frequency across follow-up periods, b.** [a] Number of participants in all studies: 0–3 months PNF n = 2128, control n = 800; 4–11 months PNF n = 2043, control n = 807; 12–23 months PNF n = 1330, control n = 316. [b] Insufficient studies were available for meta-analyses at 24 months+.

**Symptom severity: Main analyses.** As shown in Fig 3, there were significantly lower symptom severity scores in the pure PNF group versus the control at the 0–3 month follow up period, with a small effect size. This analysis included eleven studies (ten alcohol studies and one gambling study), where heterogeneity was minimal. Results were non-significant for further follow up periods of 4–11 months (k = 7) and 12–23 months (k = 2).

## Mixed PNF vs control

**Frequency: Main analyses.** As shown in Fig 4, eleven studies were included in the meta-analysis for the 0–3 month follow up period: seven alcohol studies, two gambling studies and two illicit drug use studies. The pooled SMD indicated small but significantly lower frequency in the mixed PNF group compared to the control groups, with minimal heterogeneity. At 4–11 months, seven studies (six focused on alcohol and one focused on illicit drugs) were meta-

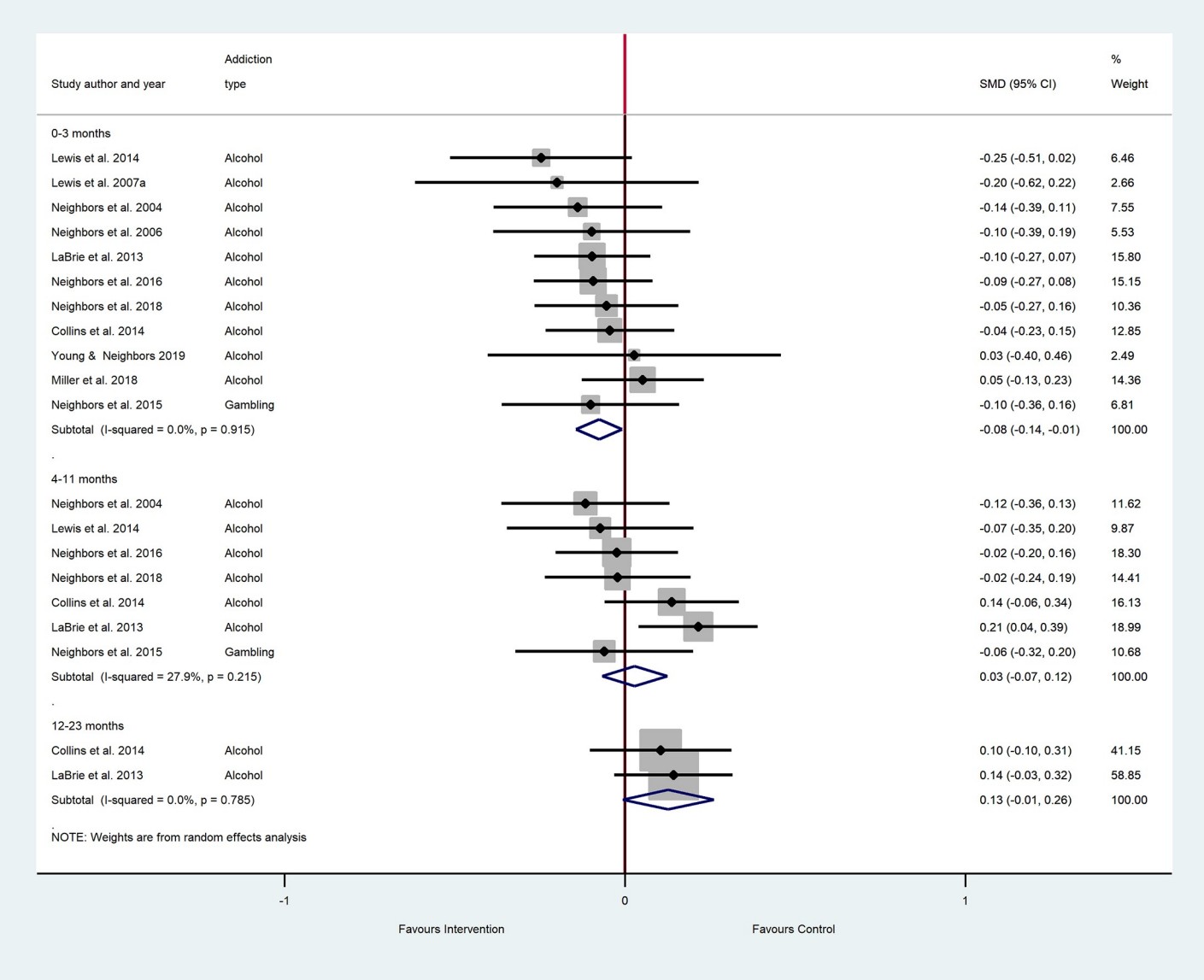

**Fig 3. The efficacy of pure PNF on symptom severity across follow-up periods[a, b].** [a] Number of participants in all studies: 0–3 months PNF n = 3396, control n = 1364; 4–11 months PNF n = 2753, control n = 950; 12–23 months PNF n = 1330, control n = 316. [b] Insufficient studies were available for meta-analyses at 24 months+.

analyzed, again showing small but significantly lower frequency in the mixed PNF group than the control groups, where heterogeneity was moderate. At 12–23 months, only two studies were available for meta-analyses, both on alcohol. This analysis showed no significant difference between mixed PNF and control groups for frequency, and heterogeneity was minimal.

**Symptom severity: Main analyses.** As shown in Fig 5, there was no significant difference in symptom severity amongst mixed PNF participants compared to controls for the 0–3 month follow-up period. Fifteen studies were available for this meta-analysis (11 alcohol studies, 2 gambling studies, and 2 illicit drug studies), and heterogeneity was minimal. Results were also non-significant for the 4–11 month follow up period, the analysis comprising 9 alcohol studies and one illicit gambling study, with substantial heterogeneity. The 12–23 month

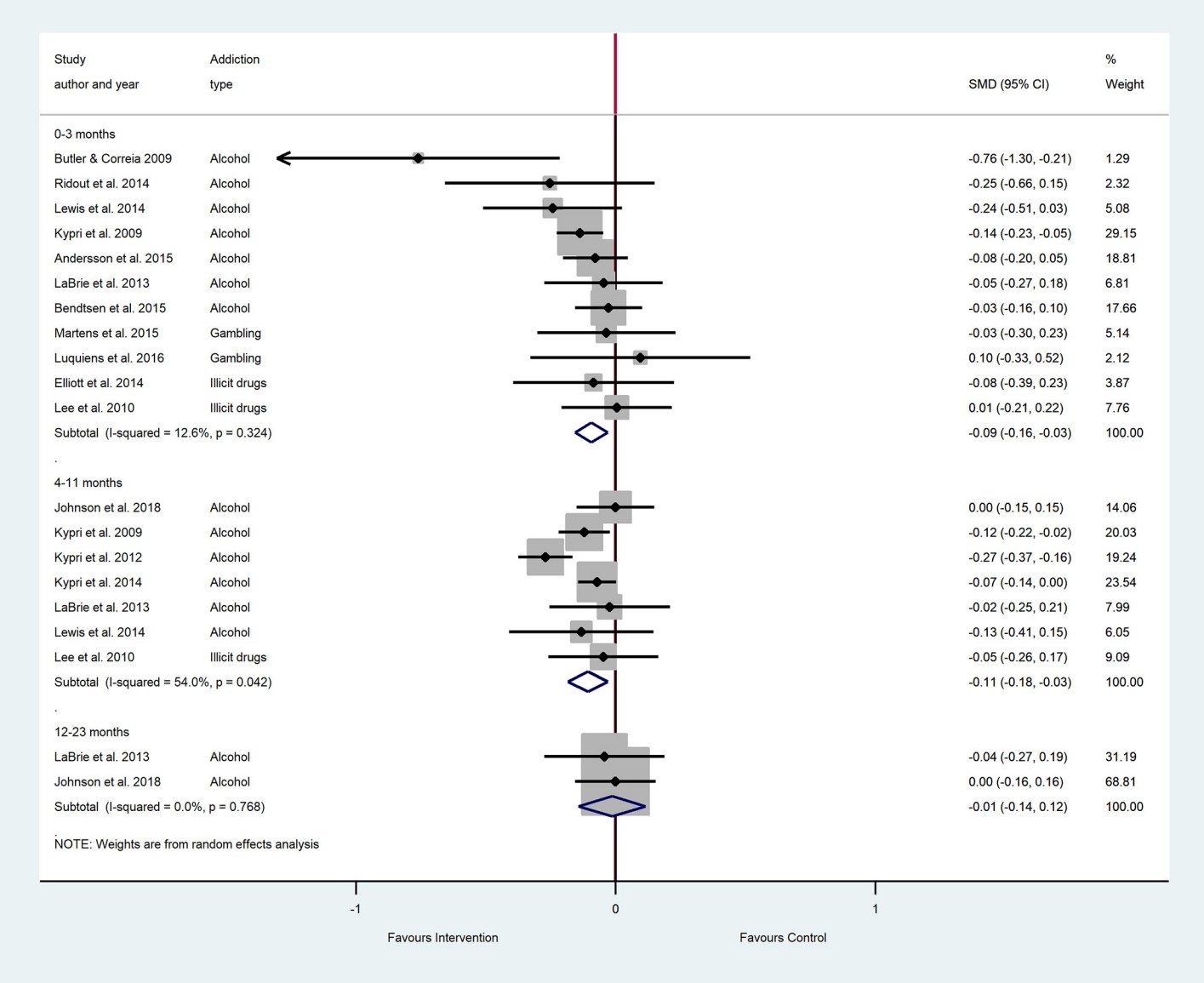

**Fig 4. The efficacy of mixed PNF on frequency across follow-up periods[a, b].** [a] Number of participants in all studies: 0–3 months PNF n = 3192, control n = 2532; 4–11 months PNF n = 3722, control n = 3639; 12–23 months PNF n = 444, control n = 478[b] Insufficient studies were available for meta-analyses at 24 months+.

analysis included 2 alcohol studies with minimal heterogeneity, and indicated a significant between group difference that favored the control group.

## Discussion

### Main findings

This is the first systematic review and meta-analysis evaluating the effectiveness of PNF interventions to address four addictive behaviors: hazardous alcohol use, problem gambling, illicit drug and tobacco use. Our review provides mixed evidence for the use of PNF to address common Substance-Related and Addictive Disorders, but there was a notable lack of studies focused on illicit drugs, few on gambling, and none on tobacco use. This limited our statistical

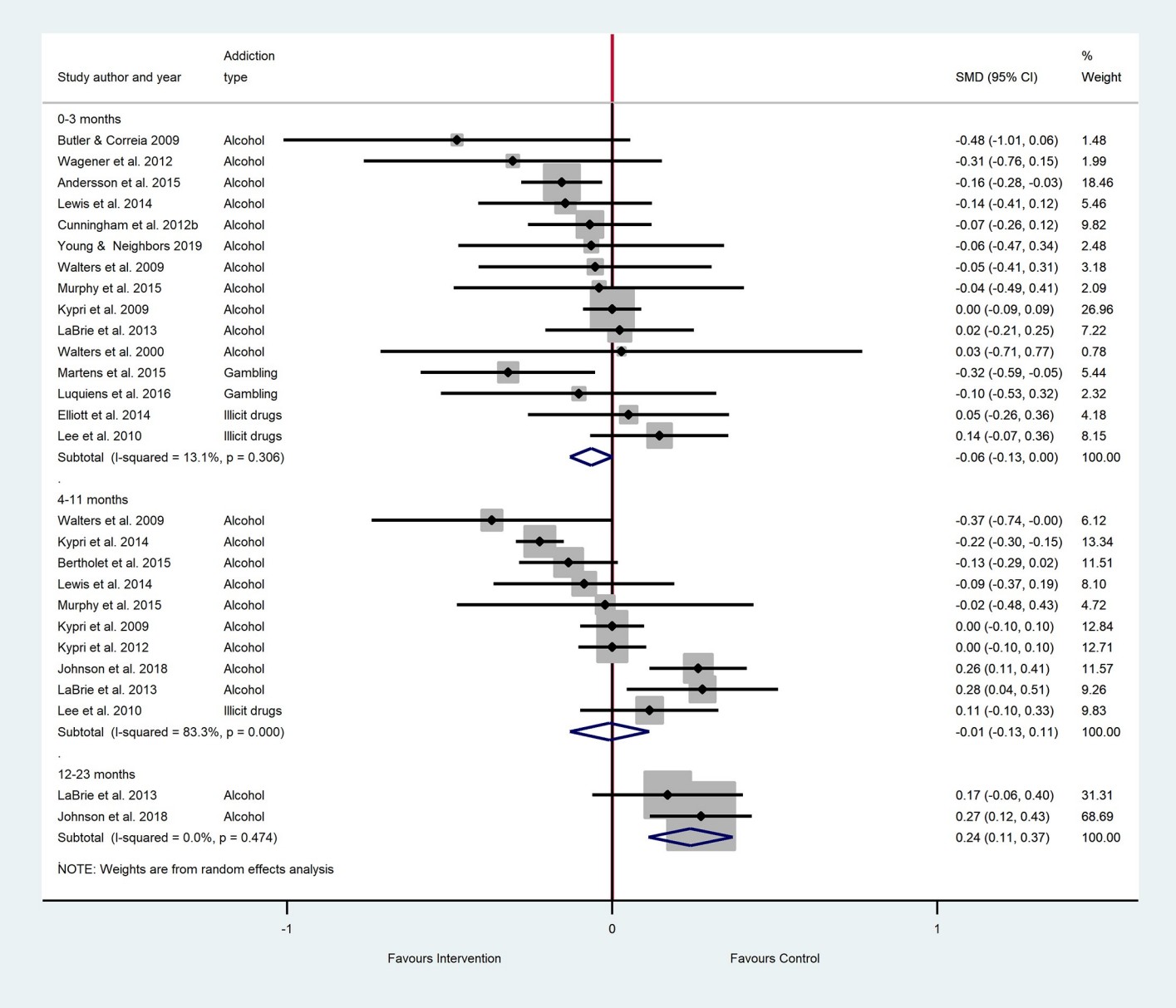

**Fig 5. The efficacy of mixed PNF on symptom severity across follow-up periods[a, b].** [a] Number of participants in all studies: 0–3 months PNF n = 3145, control n = 2369; 4–11 months PNF n = 4149, control n = 4068; 12–23 months PNF n = 444, control n = 478. [b] Insufficient studies were available for meta-analyses at 24 months+.

power and our results should be interpreted with caution. Without additional studies in these areas, we cannot draw firm conclusions about the utility of PNF across addictive behaviors. Our findings do provide some support for the use of PNF to address alcohol frequency and symptom severity in college/university settings, and to a lesser extent, gambling symptom severity in a broader range of settings. Mixed PNF studies appear to have slightly more enduring effects for some comparisons (up to 11 months) than pure PNF studies (0–3 months). With so few longer-term studies available we cannot draw firm conclusions about PNF's longevity. There is no evidence from our review that PNF can reduce frequency and symptom severity from cannabis use, and no studies were available for other illicit drugs or tobacco use.

## Comparison to the wider literature

**Alcohol.** Tanner-Smith and Lipsey [47] reported similar effect sizes to ours for frequency in two meta-analyses of interventions which overlap with PNF (norms referencing and personalized feedback), with young people 19–30 years, where their outcome (alcohol consumption) included frequency, quantity and blood alcohol content. They reported a slightly larger and more persistent effect size for symptom severity (up to one year). Their inclusion criteria did allow face-to-face contact with health professionals though, which our review did not, and which are generally associated with larger effect sizes [41]. A review by Smedslund et al. [50] of prevention studies (i.e., not yet problem users) also identified similar effect sizes to ours in short and longer-term (≥6 months) studies, though their estimates were based on low quality evidence. A review of reviews by Stockings et al. [46] reported a small reduction in problematic alcohol use (defined as heavy use, which might cause harm to self or others) amongst 10–24 year olds in response to social norms feedback, but the authors considered the size of effect to be of no meaningful benefit. Schmidt et al. [45] reported smaller effect sizes than ours, which were non-significant at short term follow up, for interventions involving printed or computer-generated feedback seeking to reduce numbers of heavy drinking episodes. Finally, Foxcroft et al. [39] reported modest reductions in alcohol frequency, with smaller effect sizes than ours, for web/computer based normative feedback interventions; they reported no effect for mailed normative feedback. Both groups of interventions permitted the inclusion of non-personalized norms, and participants who were not necessarily problem drinkers, which could explain their smaller effect sizes. Finally, a review of standalone PNF also observed small, but significant reductions in alcohol related harm, with a larger effect size than ours, though their sole focus was on college students for whom PNF seems to be the most effective [49].

**Gambling.** Our non-significant findings for gambling frequency contrast with a narrative review by Marchica and Derevensky [38], which reported promising results for personalized feedback interventions, including PNF. This difference could be explained by their inclusion of a mixture of face-to-face and self-directed PFI interventions in their analyses, whilst ours was restricted to self-directed. Meta-analyses by Goslar et al. [41] identified small significant effect sizes in the short term for self-guided treatments (CBT-based workbooks, and personalized feedback), but where effect sizes were close to zero for longer follow up periods. The greater variety of interventions (including non-PNF) they tested together could explain the difference in results compared to our review. Finally, Quilty et al. [43] identified short-term improvements in gambling behavior and associated problems in response to brief interventions (including PNF, MI/enhancement, personalized feedback and brief advice). Again, this contrasts with our finding, though the outcomes are not directly comparable, and Quilty et als gambling behavior variable included presence/absence of gambling and severity, as well as frequency.

In line with our findings for gambling symptom severity, Quilty et al. [43] detected small and significant improvements in gambling problems of a similar effect size in their meta-analysis of brief face-to-face gambling interventions. In contrast, Goslar et al. [41] identified non significant small effect sizes for global severity in post-treatment and follow up periods.

**Illicit drug use.** In line with our findings, Stockings et al. [46] concluded that social norms feedback is ineffective for addressing heavy drug use amongst people 10–24 years, based on low levels of evidence. Similarly, Smedslund et al. [50] reviewed prevention studies and observed no impact of computerized brief interventions on cannabis use in short and longer terms, as well as noting the general lack of studies for this behavior. Possible explanations for the ineffectiveness of SBIs in general for illicit drug use are offered by a narrative review, which considers illicit drug use as a different category of behavior than alcohol, for example

[44]. The author highlights that drugs are taken, despite widespread knowledge that they are (often) illegal and socially unacceptable, and users may not respond to normative information in the same way as legal and socially sanctioned behaviors such as alcohol.

## Additional intervention components

Our findings are in line with meta-analyses by Tanner-Smith and Lipsey [47], who considered various intervention components similar to the categories emerging in our papers, which were not associated with significantly larger or smaller benefits on their alcohol outcomes.

Though we did not formally compare the efficacy of pure and mixed PNF studies, both produced similar effect sizes. We did observe that mixed PNF studies reported significant findings for two medium term follow up periods (4–11 months) whereas pure PNF studies did not. It is possible that the additional components in mixed PNF studies led to more meaningful interventions (and therefore more enduring effects) from a participant perspective, but there were also far fewer pure PNF studies than mixed PNF studies, and in the absence of formal comparisons of the two types of PNF interventions, it is difficult to draw firm conclusions about whether mixed approaches are superior.

It was notable that in the PNF studies overall, very few studies (k = 1 pure, k = 2 mixed) made use of injunctive norms, so for that specific dimension, PNF interventions remain largely untested.

## Setting

College/university settings were by far the most common intervention environment. There were insufficient studies from non-college/university settings to enable equivalent subgroup analyses. In other reviews, Tanner-Smith and Lipsey [47] found that whilst university, primary health care and remote/online settings gave similar results, emergency room settings did not result in significant benefits for young adults' alcohol consumption or related problems. Conversely, Schmidt et al. [45] focused their meta-analyses on the efficacy of brief interventions (including PNF) in emergency departments on various alcohol outcomes (consumption quantity, intensity and number of heavy drinking episodes), and the majority of their comparisons indicated modest but significant effect sizes for up to 12 months. Further studies in non-university environments could help clarify the efficacy of PNF in a wider range of settings.

## Follow up period

Most comparisons that were significant in our meta-analyses showed a weakening of effects over time, though one analysis (mixed PNF studies and alcohol symptom severity) saw a short-term effect favoring the intervention group turn into a long-term effect favoring the control group. The general weakening of effects we observed is in line with other reviews [45, 47]. Weakening effects are unsurprising given the brevity of PNF, but it highlights the potential benefit of repeating interventions before 12 months for sustained behavior change, though the evidence-base for whether repeating interventions is worthwhile is currently scarce.

## Effect sizes for PNF versus other brief interventions

The modest significant effect sizes we identified for PNF appear to be similar to other brief interventions assessing frequency and symptom severity. For example, Tanner-Smith and Lipsey [47] compared the following SBIs to controls, where effect sizes are shown in parentheses for alcohol consumption and alcohol related problems respectively: CBT (0.13 and 0.10), Motivational Enhancement Therapy (0.20 and 0.17), Psycho-Educational Therapy (0.16 and 0.13).

Combining Motivational Enhancement Therapy and CBT was counterproductive (0.03 and 0.00) whilst Expectancy Challenge resulted in the strongest effect sizes (0.36 and 0.34).

## Limitations of the current evidence base

Firstly, our use of follow up means, rather than change from baseline could have over or underestimated PNF effects, depending on whether there were baseline imbalances in the outcome and which group they favored, though a recent paper suggests that this would not necessarily have changed our conclusions [92]. Several papers we included did not report baseline values, and as we were relying on published estimates without consulting authors of primary articles, we preferred to include these articles in our meta-analyses rather than omit them due to missing baseline data. As we only included RCTs, we also expected any baseline imbalances to be random, affecting intervention and control groups approximately equally. A second limitation to our review is the exclusion of non-English language articles which could have increased the number of studies included in our review. Thirdly, many of the included studies did not publish protocols prior to their trials. Since 2005, medical journals have required health related RCTs, including those that are behavioral treatments or educational programs, to be registered with an appropriate registry [93], but this has not been extended to the majority of addiction-related journals. In the absence of protocols, we relied upon the congruency of hypotheses and analysis plans with the results reported in the original articles, which could have overestimated study quality and affected our conclusions. Finally, in the context of other PFI and PNF systematic reviews, which have some degree of overlap with the present review, the impact of this paper may be incremental. However, it is also the first review to provide evidence about the efficacy of PNF across multiple addictive behaviors and settings, and we hope is useful for practitioners and users seeking to access a 'ready to go' low cost SBI.

## Implications for clinical practice

How meaningful are our effect sizes? In a review of reviews of social norms interventions to reduce risky alcohol use in young people, Stockings et al. [46], concluded that though these led to reduced alcohol use, the small associated effect sizes raise questions as to the benefit at policy and practice levels. Whilst their inclusion criteria were broader than ours and are not specific to PNF, they raise an important question about whether it is worthwhile delivering interventions with such small effects. We cautiously concur with Tanner-Smith and Lipsey [47] who address the issue of modest effects and offer a different interpretation. They conclude that such interventions were 'potentially worthwhile given their brevity and low cost', and go on to say that brief interventions are not usually intended as full treatments but as a precursor to further interventions for those who need them, as well as to motivate and provide participants with tools and resources to manage their behaviors. If adopting this perspective, future research could investigate whether PNF is a useful kick start to more intensive intervention as necessary, or as a standalone intervention for people at the lower end of the risk continuum to assist with motivation and consideration of behavior change. Schmidt et al. [45] consider that the small effect sizes they observed in emergency departments warrant a 'more cautious approach to widespread implementation' of brief interventions in those settings. They do suggest that very brief and/or computerized approaches are preferred over more resource intensive brief interventions given resource and time pressures in emergency departments, and here there is a potential place for computerized PNF in further research.

Whilst there is not strong support for implementing interventions with such small effect sizes in environments with extreme resource constraints, cost-effectiveness studies would provide useful insight as to the true value of implementing interventions, such as computerized

PNF, in a variety of settings. Available cost-effectiveness studies are scarce, and we are not aware of any assessing computerized PNF, though some studies are available for other brief interventions. One UK-based study using alcohol health workers to implement brief interventions to excessive drinkers attending sexual health clinics found mixed effects on alcohol outcomes at 6 months, with a mean cost of £12.60 per participant, which they considered was not a cost-effective use of resources [94]https://www.ncbi.nlm.nih.gov/pubmed/24813652. Conversely, one 2001 study in Australian primary health care estimated the cost of brief interventions for alcohol reduction at AUD$19.14-$21.50 and reported marginal costs per additional life year saved as below AUD$1873, which they describe as 'highly encouraging' [95]. A more recent modelling study concluded that national screening and brief intervention programs would be a cost effective way to reduce alcohol related morbidity and mortality in most EU countries [96].

The evidence is clear from longer term follow up studies that there is a time-limited effect of PNF, which largely disappears by 12 months, or sooner. This positions PNF more as a potentially useful beginning to further intensive intervention for those who need it, which could be incorporated into general health screening. PNF could be used to alert people to problem behavior, and be used to prompt motivation and engagement for addressing it, rather than PNF being a standalone intervention expected to produce sustainable change. It may be however that PNF is efficacious for alcohol than other addictions because of differing levels of importance and readiness to change. For example, people with gambling, illicit drugs and tobacco use report higher levels of perceived importance to change consumption compared to those engaged in hazardous alcohol use [97–99]. These studies indicate perceived importance of change is predictive of outcomes when combined with confidence to change. Future research should investigate the impact of SBIs that combine awareness raising and strategies for action on varying levels of readiness for change. It is unclear whether PNF booster sessions would improve the longevity of benefits. Arguably, PNF could still be used as standalone, single intervention in college/university settings to minimize harm from alcohol consumption in the short-term.

## Concluding statement

Our review provides evidence for the short-term efficacy of self-directed PNF to reduce alcohol frequency and symptom severity, and to a lesser extent gambling symptom severity based on a small number of studies. Our review does not provide evidence that self-directed PNF can be beneficial for addressing cannabis use, though again the number of available studies was very small and there were no studies of other illicit drugs. There were no studies addressing tobacco use. PNF studies tended to be conducted in college/university populations of young people, with predominantly White participants, and at the lower end of the spectrum of problem use which limits the generalizability of our findings. Whilst effects from mixed PNF studies appear to be slightly more enduring than pure PNF studies, our analyses did not provide support for the use of additional components to enhance PNF, again limited by the number of available studies. All significant effect sizes were small, but comparable with other more costly face-to-face interventions in primary care and emergency department settings. Cost effectiveness studies will help to resolve questions about whether implementing self-directed PNF at large scale is worthwhile.

## Supporting information

**S1 Checklist.**
(DOC)

**S1 Appendix. Search terms for literature search.**
(DOCX)

**S2 Appendix. Decision rules for the review and meta-analysis.**
(DOCX)

**S3 Appendix. Description of the interventions.**
(DOCX)

**S4 Appendix. Sub-group and sensitivity analyses for pure PNF studies.**
(DOCX)

**S5 Appendix. Sub-group and sensitivity analyses for mixed PNF studies.**
(DOCX)

## Acknowledgments

We acknowledge Brenna Knaebe and Matt Brittain for their valuable help with data extraction.

## Author Contributions

**Conceptualization:** Simone N. Rodda, Nicki A. Dowling.

**Data curation:** Natalia Booth.

**Formal analysis:** Jenny Saxton, Stephanie S. Merkouris.

**Funding acquisition:** Simone N. Rodda.

**Methodology:** Jenny Saxton, Simone N. Rodda, Natalia Booth, Stephanie S. Merkouris, Nicki A. Dowling.

**Project administration:** Natalia Booth.

**Supervision:** Stephanie S. Merkouris, Nicki A. Dowling.

**Writing – original draft:** Jenny Saxton, Nicki A. Dowling.

**Writing – review & editing:** Jenny Saxton, Simone N. Rodda, Natalia Booth, Stephanie S. Merkouris, Nicki A. Dowling.

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
