## [Decision Letter · Decision Letter 0]

25 Aug 2020

PONE-D-20-19468

The efficacy of Personalized Normative Feedback interventions across addictions: A systematic review and meta-analysis

PLOS ONE

Dear Dr.Rodda

Thank you for submitting your manuscript to PLOS ONE. After careful consideration, we feel that it has merit but does not fully meet PLOS ONE’s publication criteria as it currently stands. Therefore, we invite you to submit a revised version of the manuscript that addresses the points raised during the review process.

We look forward to receiving your revised manuscript.

Kind regards,

Fabio Cardoso Cruz, PhD

Academic Editor

PLOS ONE

Reviewers' comments:

Reviewer's Responses to Questions

**Comments to the Author**

1. Is the manuscript technically sound, and do the data support the conclusions?

Reviewer #1: Yes

Reviewer #2: No

2. Has the statistical analysis been performed appropriately and rigorously? 

Reviewer #1: Yes

Reviewer #2: No

3. Have the authors made all data underlying the findings in their manuscript fully available?

Reviewer #1: Yes

Reviewer #2: No

4. Is the manuscript presented in an intelligible fashion and written in standard English?

Reviewer #1: Yes

Reviewer #2: No

5. Review Comments to the Author

Reviewer #1: Saxton, Rodda and colleagues investigated through a systematic review followed by meta-analysis, the efficacy of Personalized Normative Feedback (PNF) alone, and with additional interventions (mixed PNF interventions) for hazardous alcohol use, problem gambling, illicit drugs use and smoking. Thirty four studies were included on systematic review which thirty had suitable data for meta-analysis. They found that PNF alone and mixed PNF interventions were effective in reduction short-term alcohol consumption and symptom severity. In gambling problem, mixed PNF interventions reduced short-term symptom severity. Data from illicit drugs and smoking was not enough to assess the efficacy of PNF in these cases.

The study was well designed and achieved the goals proposed by the authors, but there are some minor issues that need to be addressed to improve the manuscript.

Introduction

I missed a brief historical contextualization of PNF. You may include who proposed and validated the PNF for the first time, mainly for behavior addictions.

Review rationale

DSM-V was published in 2013 considering addiction problems as substance use and gambling. However, on page 481 there is an introduction text wherein they say “Other excessive behavioral patterns, such as Internet gaming, have also been described, but the research on these and other behavioral syndromes is less clear. Thus, groups of repetitive behaviors, which some term behavioral addictions, with such subcategories as "sex addiction," "exercise addiction," or "shopping addiction," are not included because at this time there is insufficient peer-reviewed evidence to establish the diagnostic criteria and course descriptions needed to identify these behaviors as mental disorders”.

Regarding there are several studies showing shared neurobiological mechanisms among drug use, gambling and other behavioral addictions, why did you not include this terms on systematic search? If was a personal choice I think you could include an explanation about that. Why not include other behavioral addictions?

Search strategy

Better explanation why the year interval was from 2000 to 2019.

Clarify how you conducted the systematic search.

- How did you combine the keywords?

- Why did you use different terms in different databases?

- You could improve the design of appendix 1.

Reviewer #2: By the abstract, the study aim seems interisting but tricky: to examine the efficacy of Personalized Normative Feedback (PNF) interventions upon alcohol, gambling, illicit drugs and smoking disorders and, to be widely used for all the this four disorders at the same time, once it is a low-cost intervention strategy. However the authors failed to produce an effective statistical analysis and satisfactory discussion and conclusion that could explain their results. Also, is important to be registered that the protocol and/or parts of this review was sent (but not published, according to the website) in PROSPERO database and is under review since 2018.

The text is full of gramatical problems, not appropriate words (e.g. usage of the word "hazardous" instead of alcohol use disorder, usage of the word "setting" instead of social insertion, usage of the sentence "illicit drug practices" instead of illicit drug disorders) and confused in many parts, making reading really difficult. Also, along the text the authors use the word "addiction" instead of "disorder" (or use in few moments) that is what PNFs are used to identify: disorders related to substance seeking and consumption.

The first 2 serious concerns about the paper are conceptual errors: the first is that the authors affirm in "Review Rationale" that they aimed understand if PNFs could be used as a candidate intervention for treatment of all the four disorders. However, PNFs is widely known for being a brief intervention tool that are not expected to be used as full treatment but as a personal screening tool to identify individuals that need further interventions (psychological, psychiatric) related to seeking and consumption disorders (the authors cite that in page 53 using another authors as reference, however is not possible to identify if it is also their opinion because they write shortly thereafter that PNFs could be used as "standalone intervention"); the second conceptual error is that the authors also affirm in "Review Rationale" that alcohol, gambling, illicit drugs and smoking disorders share "commom mechanisms" and used that sentence as a justification for the use of a common PNF for all this disorders. We have a problematic things in here: What the authors call as common mechanisms? The authors they used as a reference say that behavioral and substance addiction share etiological, phenomenological and clinical mechanims and the only one related to a neurobiological perspective is dopamine dysregulation. Further, the authors used as reference also affirm that there are neurobiological differences between behavioral and substance addiction (and give some examples). This (and all the literature related to drug disorders and addiction) leads us to think that an PNF (pure or mixed) would be tricky and not be effective to be used for all the four disorders discussed in this review, because there are many neurobiological, social, age, culture, sexual and several other aspects to be considered related to those disorders and addictions.

One of the aim of the authors, and that they say makes this review different from the protocol sent to PROSPERO, is that they used smoking disorders as one of the PNFs used for this disorder. However, there is any reference that was found and used in this review. Also, it was not clear why the authors used controlled prescription substances (such as sedatives, benzodiazepines and ketamine) and "designed drugs" in their search terms - there are no mentions about why they considered it and why they searched but did not mention that along the text.

One of the "Study Eligibility Criteria" was that the studies had at least one outcome (frequency and/or sympton severity). However, frequency without quantity is subjective and it was not explained why they did not use both consumtption patterns. Further, they justified that they excluded outcomes measuring quantity-type and binge-related outcomes because they could measure "different construct" - it is not clear what that means once most studies use AUDIT and other similar tools to measure the amount of drinks, for example. And in the discussion session they write about the difficulty to find other reviews that use frequency outcomes as a criteria (making it clear that studies usually prefer using quantity intead of frequency). Also, they used sympton severity as another outcome without explaining why and the criteria used to chose it as a study criteria. The authors also used 3 studies that could be argued to be selected as an exclusion criteria, as they wrote in page 9: one study with war veterans, one study in an army recruitment center and, another in outpatients waiting room; there were no explanations why there were used. Also, the authors used as an exclusion criteria people with specific physical or psychological comorbidities, however they did not explained if the studies they used all the participants had or had not comorbidities (one aspect that is super common in individuals with drug-related disorders).

There are no diversity among the populations, nationality and ages in the studies used as reference. In "Pure PNF" tables is used only studies from the USA and in female young University population. And in "Mixed PNF" tables also most studies are from USA and in young University population. In most those studies there are only one outcome (frequency or severity), in the majority there are no effective results and, basically all the studies are alcohol-related disorders papers (among the 37 papers used as references, only 3 are about gambling and other 3 with marijuana - and not illicit drugs as the authors used to refer it). However, the authors insisted along the results, discussion and conclusion that "...it is also the first review to provide evidence about the efficacy of PNF across multiple addictive behaviours and settings..." (page 51). The same authors used the sentence "...but there was a lack of studies focused on illicit drugs, few on gambling and, none on smoking which limited our statistical power. Without additional studies in these areas, we cannot draw firm conclusions about the utility of PNF across addictive behaviors." (page 44). Also, in page 54 the authors are not conclusive about their opinion if PNF could be or could be not be used, as they propose, "a standalone intervention". Considering those conflicting sentences and the statistical methods used (explained below), it is clear that the authors are not sure about their results and conclusions and, is also clear that the complexity of the drugs abuse disorders and all the other factors involved is not under consideration in this review.

About the statistical analyses, in the beggining of the text the authors write that they converted medians to "means" (average) but it is not clear why. Also, in page 18 (item h) the authors write "Means and SDs were not extractable from the paper due to poor resolution of graphs", this leaded me to ask: How did they collect all the other results? Did they collect all the data used in this review from graphics?. If so, this compromises all the review results, discussion and conclusion. Also, in almost all the statistical analyses they removed studies (most of them the ones with no effect or no results about the frequency and/or severity) and this also compromises this review. The question in here is: Why not using the data the are available in the papers used as reference?.

Finally, in page 51 the authors write "Studies of other illicit drugs are notably absent, and researchers continue to be challenged by recruitment and other practical obstacles to adressing behaviors that participants know are agains the law". Behaviors related to drug use, abuse and addiction are not against the law, drug traffic is against the law. Drug abuse disorders and drug addiction is a serious world health problem and individuals that suffer with it must be treated with empathy and kindness once it is a disease just like any other and treatment is necessary. Most of them are marginalized populations with no access to doctors and multidisciplinary strategies to treat their illness. Most of them do not know that their condition is a disease because the sociaty still stigmatizes their illness as a character flaw. Participants maybe do not easily share their disorders related to drugs use/abuse because they are treated as outlaw persons and not as what they really are: people who need adequate treatment and attention.

6. PLOS authors have the option to publish the peer review history of their article (what does this mean?). If published, this will include your full peer review and any attached files.

Reviewer #1: No

Reviewer #2: No

---

## [Author Response · Author response to Decision Letter 0]

1 Oct 2020

Reviewer #1: Saxton, Rodda and colleagues investigated through a systematic review followed by meta-analysis, the efficacy of Personalized Normative Feedback (PNF) alone, and with additional interventions (mixed PNF interventions) for hazardous alcohol use, problem gambling, illicit drugs use and smoking. Thirty four studies were included on systematic review which thirty had suitable data for meta-analysis. They found that PNF alone and mixed PNF interventions were effective in reduction short-term alcohol consumption and symptom severity. In gambling problem, mixed PNF interventions reduced short-term symptom severity. Data from illicit drugs and smoking was not enough to assess the efficacy of PNF in these cases. The study was well designed and achieved the goals proposed by the authors, but there are some minor issues that need to be addressed to improve the manuscript.

Thank you to the reviewer for their positive response to the manuscript.

I missed a brief historical contextualization of PNF. You may include who proposed and validated the PNF for the first time, mainly for behavior addictions.

- Regarding the historical context we have now included a paragraph on the origins of PNF for addictive behaviours.

Review rationale: DSM-V was published in 2013 considering addiction problems as substance use and gambling. However, on page 481 there is an introduction text wherein they say “Other excessive behavioral patterns, such as Internet gaming, have also been described, but the research on these and other behavioral syndromes is less clear. Thus, groups of repetitive behaviors, which some term behavioral addictions, with such subcategories as "sex addiction," "exercise addiction," or "shopping addiction," are not included because at this time there is insufficient peer-reviewed evidence to establish the diagnostic criteria and course descriptions needed to identify these behaviors as mental disorders”. Regarding there are several studies showing shared neurobiological mechanisms among drug use, gambling and other behavioral addictions, why did you not include this terms on systematic search? If was a personal choice I think you could include an explanation about that. Why not include other behavioral addictions?

- We agree with the reviewer that other addictions would be very interesting. PNF requires re-presentation of population norms and given the relative newness of other behavioural addictions we did not expect to find any studies. This meant we decided to include only those disorders currently classified in the DSM -5. We acknowledge, however, that future iterations of this review should include the newer behavioural addictions.

Search strategy: Better explanation why the year interval was from 2000 to 2019.

- By adding the brief history as suggested by the reviewer we believe we have now addressed the timeframe. We have added this rationale to the search strategy.

Clarify how you conducted the systematic search. How did you combine the keywords? Why did you use different terms in different databases? You could improve the design of appendix 1.

-We have edited the search strategy in text (lines 188-190) for clarity. We believe this explains the combination of keywords. Re the use of different terms in different databases, we understand that mesh terms differ across different databases. We have, where possible, maintained the same keywords for each search.

Reviewer #2: By the abstract, the study aim seems interisting but tricky: to examine the efficacy of Personalized Normative Feedback (PNF) interventions upon alcohol, gambling, illicit drugs and smoking disorders and, to be widely used for all the this four disorders at the same time, once it is a low-cost intervention strategy. However the authors failed to produce an effective statistical analysis and satisfactory discussion and conclusion that could explain their results. Also, is important to be registered that the protocol and/or parts of this review was sent (but not published, according to the website) in PROSPERO database and is under review since 2018.

- The reviewer notes that the review was registered in 2018. We believe this is not an unusual amount of time to undertake a complex review. Moreover, the search was updated prior to submission with PLOS One. With regards to the comment that we are suggesting PNF is used for all four disorders at the same time – this was not suggested in the article. Rather, we are arguing that this type of brief intervention could be used for all disorders classified in the DSM-5 as addictions. With reference to the results, we address this below where the reviewer has more fully expressed their concerns.

The text is full of gramatical problems, not appropriate words (e.g. usage of the word "hazardous" instead of alcohol use disorder, usage of the word "setting" instead of social insertion, usage of the sentence "illicit drug practices" instead of illicit drug disorders) and confused in many parts, making reading really difficult. Also, along the text the authors use the word "addiction" instead of "disorder" (or use in few moments) that is what PNFs are used to identify: disorders related to substance seeking and consumption.

-We have reviewed the manuscript to ensure wording is consistent throughout. In particular, we have removed the words ‘practices’, addiction (where less appropriate). However, our eligibility criteria included study samples that consisted of individuals with some level of problematic use of alcohol, gambling, drugs, or tobacco use at baseline. 

The term “hazardous use” has therefore been retained as this is a term used in the PNF literature to refer to problematic levels of alcohol consumption that do not necessarily constitute alcohol use disorder (see reference list for example).

The first 2 serious concerns about the paper are conceptual errors: the first is that the authors affirm in "Review Rationale" that they aimed understand if PNFs could be used as a candidate intervention for treatment of all the four disorders. However, PNFs is widely known for being a brief intervention tool that are not expected to be used as full treatment but as a personal screening tool to identify individuals that need further interventions (psychological, psychiatric) related to seeking and consumption disorders (the authors cite that in page 53 using another authors as reference, however is not possible to identify if it is also their opinion because they write shortly thereafter that PNFs could be used as "standalone intervention"); 

- We have removed reference to ‘treatment’ and retained intervention so as to differentiate PNF as a brief intervention from more intensive treatment approaches. We offer different ways PNF could be used and as stated by the reviewer, this could be as a means of identifying individuals that may be interested in face-to-face treatment. However, as indicated in multiple studies and our review, PNF can also have an impact of frequency of consumption and severity of symptoms as a standalone intervention. The wider literature and evidence in our review does not support the reviewers assertion that PNF should primarily be used for screening and referral only. We strongly disagree that this is a conceptual error and we have presented multiple perspectives on how PNF might be applied.

The second conceptual error is that the authors also affirm in "Review Rationale" that alcohol, gambling, illicit drugs and smoking disorders share "commom mechanisms" and used that sentence as a justification for the use of a common PNF for all this disorders. We have a problematic things in here: What the authors call as common mechanisms? The authors they used as a reference say that behavioral and substance addiction share etiological, phenomenological and clinical mechanims and the only one related to a neurobiological perspective is dopamine dysregulation. Further, the authors used as reference also affirm that there are neurobiological differences between behavioral and substance addiction (and give some examples). This (and all the literature related to drug disorders and addiction) leads us to think that an PNF (pure or mixed) would be tricky and not be effective to be used for all the four disorders discussed in this review, because there are many neurobiological, social, age, culture, sexual and several other aspects to be considered related to those disorders and addictions.

- We are not arguing that there is a common tool for all disorders, rather that PNF as an intervention type could be used for all addictive disorders. The actual intervention items and norms should be based on the specific area of interest. Moreover, the aim of the study is to identify whether indeed PNF can be employed successfully for each of these addictive behaviours and our analyses allow us to draw conclusions on the possible differential efficacy of PNF on these behaviours.

One of the aim of the authors, and that they say makes this review different from the protocol sent to PROSPERO, is that they used smoking disorders as one of the PNFs used for this disorder. However, there is any reference that was found and used in this review. 

- The reviewer is correct, we are now including tobacco use in our group of addictive behaviours. To our surprise, we did not locate any studies that used pure or supported PNF. Given it was in our search, it is not appropriate to remove it for publication. Moreover, the gap identified in this field of research is an important outcome of this study.

Also, it was not clear why the authors used controlled prescription substances (such as sedatives, benzodiazepines and ketamine) and "designed drugs" in their search terms - there are no mentions about why they considered it and why they searched but did not mention that along the text.

- The review was inclusive of all illicit and prescription drugs and therefore these substances were included.

One of the "Study Eligibility Criteria" was that the studies had at least one outcome (frequency and/or sympton severity). However, frequency without quantity is subjective and it was not explained why they did not use both consumtption patterns. 

- When examining multiple substance and behavioural addictions, it is not possible to consistently identify an index of quantity. This is because quantity in some addictive behaviours (e.g., gambling – where expenditure could be the quantity index) means quite a different thing to other addictive behaviours (e.g., alcohol – where number of standard drinks may be the quantity index). We therefore selected frequency and symptom severity as the key outcomes. We have included this statement in the methods- study eligibility section.

Further, they justified that they excluded outcomes measuring quantity-type and binge-related outcomes because they could measure "different construct" - it is not clear what that means once most studies use AUDIT and other similar tools to measure the amount of drinks, for example. 

- Excluded studies that examined binge-related outcomes may measure frequency but this is often in terms of frequency of binges rather than frequency of use. Moreover, data on binges and quantity (as above) do not consistently apply over all substance use and behavioural addictions

And in the discussion session they write about the difficulty to find other reviews that use frequency outcomes as a criteria (making it clear that studies usually prefer using quantity intead of frequency).

- The statement regarding limited reviews examining frequency was specific to gambling (line 867). The lack of other reviews which included frequency was due to gambling focusing more on expenditure and severity. Expenditure is not a viable comparison in our study where we included multiple substance and other additions. We also noted this requirement as a limitation on line 973.

Also, they used sympton severity as another outcome without explaining why and the criteria used to chose it as a study criteria. 

- As noted above we have now included a sentence in the methods section stating reasons for the use of frequency over quantity. That is, we selected frequency and severity as they are indices that can consistently be applied across all of the included substance and behavioural addictions.

The authors also used 3 studies that could be argued to be selected as an exclusion criteria, as they wrote in page 9: one study with war veterans, one study in an army recruitment center and, another in outpatients waiting room; there were no explanations why there were used. 

- The exclusion criteria as stated on page 260 was “PNF intervention targeted people with specific physical or psychological comorbidities (e.g., war veterans with post-traumatic stress disorder), which are less generalizable to other populations”. This is not exclusive of studies that involved veterans rather those with physical or psychological comorbidities.

Also, the authors used as an exclusion criteria people with specific physical or psychological comorbidities, however they did not explained if the studies they used all the participants had or had not comorbidities (one aspect that is super common in individuals with drug-related disorders).

- As stated above, they were excluded where the intervention targeted this group. We did not exclude intervention who reported psychical or psychological comorbidities in the participant characteristics because we agree that these comorbidities are highly relevant to the examination of these disorders.

There are no diversity among the populations, nationality and ages in the studies used as reference. In "Pure PNF" tables is used only studies from the USA and in female young University population. And in "Mixed PNF" tables also most studies are from USA and in young University population. In most those studies there are only one outcome (frequency or severity), in the majority there are no effective results and, basically all the studies are alcohol-related disorders papers (among the 37 papers used as references, only 3 are about gambling and other 3 with marijuana - and not illicit drugs as the authors used to refer it). However, the authors insisted along the results, discussion and conclusion that "...it is also the first review to provide evidence about the efficacy of PNF across multiple addictive behaviours and settings..." (page 51). 

- As noted by the reviewer, we were answering a set of a priori questions that were published on Prospero. The evidence that we provide in this review does not just relate to the efficacy of PNF, but also allows us to note an absence of empirical evidence, across the four substance and addictive disorders. In emerging fields, such as gambling, it is important for reviews to identify gaps in the evidence base, thus calling for the need for researchers to redress these gaps.

The same authors used the sentence "...but there was a lack of studies focused on illicit drugs, few on gambling and, none on smoking which limited our statistical power. Without additional studies in these areas, we cannot draw firm conclusions about the utility of PNF across addictive behaviors." (page 44). 

- As above, we were very careful not to speak beyond the regarding the efficacy of PNF for these conditions. As stated by the reviewer, the research identified a significant gap in this area and we have highlighted this gap for the consideration of future research.

Also, in page 54 the authors are not conclusive about their opinion if PNF could be or could be not be used, as they propose, "a standalone intervention". Considering those conflicting sentences and the statistical methods used (explained below), it is clear that the authors are not sure about their results and conclusions and, is also clear that the complexity of the drugs abuse disorders and all the other factors involved is not under consideration in this review.

- We have stated “PNF could be a useful kick start to more intensive intervention as necessary, or as a standalone intervention for people at the lower end of the risk continuum to get people motivated and engaged to change their practices.” We have now added a statement in the discussion that future research is required to investigate this possibility.

About the statistical analyses, in the beggining of the text the authors write that they converted medians to "means" (average) but it is not clear why. 

- Where studies only provided medians, we converted these medians to means so as to allow the meta-analysis to be conducted (see page 13 of the manuscript). As indicated in the manuscript, this is not yet standard practice for systematic reviews, but it is an acceptable approach based on the Cochrane Handbook for Systematic Reviews of 

Interventions (Higgins & Deeks, 2011) (see Higgins JPT, Deeks JJ (editors). Chapter 7: Selecting studies and collecting data. In: Higgins JPT, Green S (editors), Cochrane Handbook for Systematic Reviews of Interventions Version 5.1.0 (updated March 2011). The Cochrane Collaboration, 2011. Available from www.handbook.cochrane.org. 

Also, in page 18 (item h) the authors write "Means and SDs were not extractable from the paper due to poor resolution of graphs", this leaded me to ask: How did they collect all the other results? Did they collect all the data used in this review from graphics? If so, this compromises all the review results, discussion and conclusion. 

-Data was sourced and extracted from text and tables and graphical illustrations in rare cases (this is noted on page 13). In Table 1, we noted for this specific study (reference number 68), there was no outcome data reported in the text or table and the graph could not be interpreted. 

Also, in almost all the statistical analyses they removed studies (most of them the ones with no effect or no results about the frequency and/or severity) and this also compromises this review. The question in here is: Why not using the data the are available in the papers used as reference?.

- A meta-analysis requires data in order to be conducted. As stated by the reviewer, these excluded papers did not report frequency or severity outcomes (often descriptive papers only). 

Finally, in page 51 the authors write "Studies of other illicit drugs are notably absent, and researchers continue to be challenged by recruitment and other practical obstacles to adressing behaviors that participants know are agains the law". Behaviors related to drug use, abuse and addiction are not against the law, drug traffic is against the law. Drug abuse disorders and drug addiction is a serious world health problem and individuals that suffer with it must be treated with empathy and kindness once it is a disease just like any other and treatment is necessary. Most of them are marginalized populations with no access to doctors and multidisciplinary strategies to treat their illness. Most of them do not know that their condition is a disease because the sociaty still stigmatizes their illness as a character flaw. Participants maybe do not easily share their disorders related to drugs use/abuse because they are treated as outlaw persons and not as what they really are: people who need adequate treatment and attention.

- We do not disagree with the reviewer about the conceptualisation of these behaviours as a public health issue. We were simply highlighting a possible methodological reason for a lack of studies in this area. However, to acknowledge the reviewer’s comment, we have removed the statement related to legal issues associated with illicit drug use.

---

## [Decision Letter · Decision Letter 1]

11 Nov 2020

PONE-D-20-19468R1

The efficacy of Personalized Normative Feedback interventions across addictions: A systematic review and meta-analysis

PLOS ONE

Dear Dr.Rodda

Thank you for submitting your manuscript to PLOS ONE. After careful consideration, we feel that it has merit but does not fully meet PLOS ONE’s publication criteria as it currently stands. Therefore, we invite you to submit a revised version of the manuscript that addresses the points raised during the review process.

We look forward to receiving your revised manuscript.

Kind regards,

Fabio Cardoso Cruz, PhD

Academic Editor

PLOS ONE

Reviewers' comments:

Reviewer's Responses to Questions

**Comments to the Author**

1. If the authors have adequately addressed your comments raised in a previous round of review and you feel that this manuscript is now acceptable for publication, you may indicate that here to bypass the “Comments to the Author” section, enter your conflict of interest statement in the “Confidential to Editor” section, and submit your "Accept" recommendation.

Reviewer #1: All comments have been addressed

Reviewer #2: (No Response)

Reviewer #3: (No Response)

Reviewer #4: All comments have been addressed

2. Is the manuscript technically sound, and do the data support the conclusions?

Reviewer #1: Yes

Reviewer #2: No

Reviewer #3: Yes

Reviewer #4: Yes

3. Has the statistical analysis been performed appropriately and rigorously? 

Reviewer #1: Yes

Reviewer #2: No

Reviewer #3: Yes

Reviewer #4: Yes

4. Have the authors made all data underlying the findings in their manuscript fully available?

Reviewer #1: Yes

Reviewer #2: Yes

Reviewer #3: Yes

Reviewer #4: Yes

5. Is the manuscript presented in an intelligible fashion and written in standard English?

Reviewer #1: Yes

Reviewer #2: Yes

Reviewer #3: Yes

Reviewer #4: Yes

6. Review Comments to the Author

Reviewer #1: Saxton, Rodda and colleagues re-submitted the paper about systematic review followed by meta-analysis of the efficacy of Personalized Normative Feedback (PNF) alone, and with additional interventions (mixed PNF interventions) for addictive behaviors. Authors provided good answers for concerns highlighted by the reviewers, but there isa minor issue that needs to be addressed to improve the manuscript.

In introduction, the authors affirm, based on literature, that all addictive behaviors share a common mechanism. In Results they found differences in effects of pure and mixed PNF interventions for alcohol and gambling, but no influence for illicit drugs and tobacco use. I think authors could suggest why these differences appeared between each addictive behavior.

Reviewer #2: Despite the reviewers questions and sugestions made, the authors realized just few changes along the text and did not responded in a directed manner and/or were vague in their responses. I consider that the material that was presented and, is now once again presented without big changes, is not suitable for publication in PlosOne at this moment. Below I list point by point the reason of my final decision reviewing this paper.

#1 - The first reviewer asked for a historical contextualization of PNF (including who proposed and validated). The authors answered they included a paragraph in the text, however this paragraph is consuding and does not clearly explain what was sugested and the importance of PNF (why it was first proposed, for example).

#2 - The first reviewer asked for the authors to clarify how the systematic search was conducted and also asked for improvements in Appendix 1. The authors answered they edited the search strategy, however nothing was noticed in the "new" text and no improvements were made about the Apendix 1 design.

#3 - I mentioned that aparently the authors suggested that PNF was proposed in this review as a tool intervention for all the fours disorders at the same time. The authors answered they did not suggest that along the text. Considering that, the authors should better look at the lines 107-110 and 125-131 and rewrite the sentences to better clarify what is proposed by them.

#4 - I suggested a gramatical problem along the text and suggested few words to be changed for more appropiate words. The authors answered they reviewed the text and removed the word "practices". Indeed the authors changed the word "practices" for more appropiate words, however the text does not seem that passed throug a gramatical review. The text is still confusing along the review.

#5 - I commented about what makes this review presented to PlosOne different from the one submitted to PROSPERO (the use of smoking studies) and the absence of references about smoking studies in this review. The authors answered they included tobacco use in their groups of addictive behaviours and that they did not find any new studies, just like in the smoking related studies they did before. However, the authors did not present a new Appendix and it seems they just switched the work "smoking" for "tobbaco" along the text with no justification.

#6 - I highlighted that the authors were not clear in the text about the use of controlled prescription substances and "designed drugs" in their search. This question remains unsolved, the authors did not anwered the question and did not include anything in the text that could clearfy this information.

#7 - I asked why the authors chose frequency and/or severity instead of quantity. The authors answered explained their option for using frequency and severity and included it in the text. However, the authors affirmed those are powerfull tools without using any reference that could confirm that the use of frequencyand/or severity is more suitable than quantity.

#8 - Along the responses to the questions made, the authors demonstrated that the aim of the review would be demonstrating "the importance of reviews to idenfity gaps in the evidence base, thus calling for the need for researchers to redress these gaps". This is what the authors should have proposed once their results are basically related to it. However, the authors affirm along the text that this review presented confirm the importance of PNF despite the gaps found, and sometimes affirming its efficacy taking in consideration just one study (for gambling). I still sugest to the authors to review their aims for this review, I believe those aims clould be changed to a better result and better discussion.

#9 - I stated that the the authors opted to convert the medians of the studies used into medians, and this choose was not clear. The authors answered that they did it to allow the meta-analysis. Moreover, the authors answered that this is not the standard practice for systematic reviews but acceptable according to their reference, without explain why they chose this option instead of the standard practice for systematic reviews.

#10 - I noticed that the authors used data derivative from graphics and this was one of the points I highlighted, hoping it was just a misunderstanding. However, the authors affirmed my suspicions answering that in fact they extracted some data from graphical illustrations in rare cases. In my point of view, this is a wrong way to collect data and may compromise the final results.

#11 - Finally I had highlighted a sentence that was completely wrong and conceptually misunderstood - "Studies of other illicit drugs are notably absent, and researchers continue to be challenged by recruitment and other practical obstacles to adressing behaviours that participants know are against the law". I spent time explained in few lines how wrong this sentence is, but it looks like that the authors do not realize how stigmatized and wrong this sentence is once they changed this sentence for "Studies of other illicit drugs are notably absent, and researchers continue to be challenged by recruitment and other practical obstacles to addressing illicit behaviors". I will not extend myself explaining it again, it was well explained the other time, with no understanding by the authors. My concern is that this sentence was not a wrong way to express the gap in literature relating PNF and illicit drugs, but that this is the authors opinion about people with drug-related problems.

Reviewer #3: The systematic review and meta-analysis by Saxton, Rodda and colleagues assessed the efficacy of 'pure' personalized normative feedback (PNF) interventions and in combination with additional self-directed interventions (‘mixed’ PNF) on frequency and symptom severity of hazardous use of alcohol, tobacco, illicit drugs and problem gambling. Authors reported significant effects of 'pure' and ‘mixed’ PNF in reducing short-term frequency and symptom severity associated with alcohol use. In addition, they observed that ‘mixed’ PNF reduced gambling symptom severity in the short term. No effect of the intervention was found on outcomes associated with illicit drugs use. No tobacco study met the inclusion criteria. Authors discussed the importance of conducting future studies to examine whether PNF could be a useful kick start tool to more intensive intervention, or a standalone intervention in some cases. Authors also discussed the need for future cost-effectiveness research to evaluate whether application of PNF in a more comprehensive manner is worthwhile.

I consider that the manuscript is well written, the aims are well defined, methodology is clear and conclusions are supported by data. Authors appropriately discussed the strengths and limitations of the study.

I believe that the previous review brought improvements to the manuscript.

I would like to point out some minor issues and suggestions:

1. Please, include in the abstract what the acronym “RCT” means (= randomized controlled trial).

2. In the abstract, authors conclude, “Our findings highlight the efficacy of PNF to address alcohol frequency and symptom severity”, but what about gambling, since they affirmed “PNF with additional interventions reduced short-term gambling symptom severity”. I understand that authors did not draw a firm conclusion about the effect of PNF on gambling symptom severity due to the reduced number of gambling studies included in the analyses, but I think that this could be explained in the abstract.

3. Suggestions: on page 3 (introduction), authors could maybe introduce drugs of abuse topics first (alcohol, tobacco, illicit drugs), and then introduce gambling-related topic.

In addition, on page 10, I think authors could place the sentence “We selected frequency and severity as they are indices that can consistently be applied across all of the included substance and behavioral addictions” after mentioning both evaluated outcomes, frequency and symptom severity (at the end of the paragraph).

4. On page 10, authors could include a brief explanation about why exactly they decided to exclude quantity measures, BAC, attitudinal change and dollars spent on gambling.

5. I think that table 5 (page 38) may have formatting issues.

6. In the “Implications for clinical practice” section (discussion), authors discuss about whether PNF effects in reducing alcohol frequency and symptom severity are meaningful or not, due to the small effect sizes observed. About this, authors brought considerations that were addressed by other reviews. However, it was not clear to me the authors’ opinion on this.

7. In the “Follow up period” section (discussion, page 52), it would be nice to discuss more about the long-term effect of mixed PNF on alcohol symptom severity favoring the control group. Did authors hypothesize why this happens?

Reviewer #4: Saxton and colleagues report a systematic review and meta-analysis of the efficacy of Personalized Normative Feedback (PNF) interventions across addictions in meticulous detail based on about 30 studies. This revised manuscript represents a carefully conducted study on an interesting, albeit rather weak intervention to address a variety of addictions. The authors were

responsive to the comments from previous reviewers and made necessary changes, while avoiding recommendations that were inconsistent with their registration of the study in Prospero.

The write up of this study was very complete (~65 pages plus supplemental material), but tedious to read. It seemed that many paragraphs were essentially redundant with the substitution of another analysis or subanalysis with slightly different statistical results. In contrast to Tables 3-6, I found that the forest plot figures in the Supplemental Material (Appendices D and E) really told the whole story. They present the meta-analytic results clearly and much more succinctly than the pages of narrative that were highly redundant with the Tables. Although the figures would need to be redone to save space, the graphic depiction of results is rather compelling, negating the need for the repetitive

statistical results format presented.

The discussion was a rather lengthy recap of results, some of which did not need to be repeated. After focusing on effect sizes, I was anxious to read the authors’ interpretation of the extent to which PNF produced clinically- or societally-meaningful improvements in health and wellbeing. I felt the discussion pertained more to the limitations of the available research, or lack thereof, than the intervention itself. What I took away was that perhaps this may be cost-effective because the small and short-term effects (never clearly defined) are offset by the low cost of the intervention.

In summary, I believe this review is worthy of publication. This is a carefully done review and meta-analysis and it more than adequately summarizes what is known about the efficacy of Personalized Normative Feedback interventions. However, I am not sure that it warrants such a lengthy treatise

7. PLOS authors have the option to publish the peer review history of their article (what does this mean?). If published, this will include your full peer review and any attached files.

Reviewer #1: No

Reviewer #2: No

Reviewer #3: No

Reviewer #4: **Yes: **Howard Goldstein

---

## [Author Response · Author response to Decision Letter 1]

22 Dec 2020

6. Review Comments to the Author

Reviewer #1: Saxton, Rodda and colleagues re-submitted the paper about systematic review followed by meta-analysis of the efficacy of Personalized Normative Feedback (PNF) alone, and with additional interventions (mixed PNF interventions) for addictive behaviors. Authors provided good answers for concerns highlighted by the reviewers, but there isa minor issue that needs to be addressed to improve the manuscript.

Thank you for the positive feedback

In introduction, the authors affirm, based on literature, that all addictive behaviors share a common mechanism. In Results they found differences in effects of pure and mixed PNF interventions for alcohol and gambling, but no influence for illicit drugs and tobacco use. I think authors could suggest why these differences appeared between each addictive behavior.

We have now added a section into the implications section on the role of importance and readiness across different addictive behaviours that may, in part, explain these findings.

Reviewer #2: Despite the reviewers questions and sugestions made, the authors realized just few changes along the text and did not responded in a directed manner and/or were vague in their responses. I consider that the material that was presented and, is now once again presented without big changes, is not suitable for publication in PlosOne at this moment. Below I list point by point the reason of my final decision reviewing this paper.

Thank you Reviewer 2 for the time they have spent on our manuscript. We have carefully reviewed their first and second round of comments and strongly refute the claim that without big changes the manuscript is not suitable for publication. The PLOS One guidelines state peer review is used “to determine whether a paper is technically rigorous and meets the scientific and ethical standard for inclusion in the published scientific record.” We believe that we have met and exceeded these standards as indicated by three of the four reviewers. We have ensured the work is aligned with established standards relevant to systematic reviews including PRISMA and Cochrane. Our study was also registered with PROSPERO prior to its undertaking and we have undertaken our analysis consistent with the registered protocol. 

#1 - The first reviewer asked for a historical contextualization of PNF (including who proposed and validated). The authors answered they included a paragraph in the text, however this paragraph is consuding and does not clearly explain what was sugested and the importance of PNF (why it was first proposed, for example).

As indicated by Reviewer 1, the historical contextualisation of PNF that we provided was satisfactory. As stated in the text, PNF was first developed in the US in response to college student drinking. 

#2 - The first reviewer asked for the authors to clarify how the systematic search was conducted and also asked for improvements in Appendix 1. The authors answered they edited the search strategy, however nothing was noticed in the "new" text and no improvements were made about the Apendix 1 design.

Apologies for the confusion. These changes were made in the first revision in response to Reviewer 1 but the relevant changes were not highlighted in the manuscript. 

#3 - I mentioned that aparently the authors suggested that PNF was proposed in this review as a tool intervention for all the fours disorders at the same time. The authors answered they did not suggest that along the text. Considering that, the authors should better look at the lines 107-110 and 125-131 and rewrite the sentences to better clarify what is proposed by them.

Thank you we have clarified that the intention is to look at the use of PNF for each of these different addictive disorders – not as one treatment.

#4 - I suggested a gramatical problem along the text and suggested few words to be changed for more appropiate words. The authors answered they reviewed the text and removed the word "practices". Indeed the authors changed the word "practices" for more appropiate words, however the text does not seem that passed throug a gramatical review. The text is still confusing along the review.

The text has been reviewed and any grammatical errors have been amended. 

#5 - I commented about what makes this review presented to PlosOne different from the one submitted to PROSPERO (the use of smoking studies) and the absence of references about smoking studies in this review. The authors answered they included tobacco use in their groups of addictive behaviours and that they did not find any new studies, just like in the smoking related studies they did before. However, the authors did not present a new Appendix and it seems they just switched the work "smoking" for "tobbaco" along the text with no justification.

The reviewer is correct that the Appendix reflects the search which was conducted as stated. As noted in the search terms both tobacco and smoking were used. Changes in the text were in response to the reviewers request for consistency in terms.

#6 - I highlighted that the authors were not clear in the text about the use of controlled prescription substances and "designed drugs" in their search. This question remains unsolved, the authors did not anwered the question and did not include anything in the text that could clearfy this information.

The Study Eligibility Criteria section of the manuscript has been edited to clearly indicate that the review was inclusive of all illicit and prescription drugs. The search terms are not able to be changed now. 

#7 - I asked why the authors chose frequency and/or severity instead of quantity. The authors answered explained their option for using frequency and severity and included it in the text. However, the authors affirmed those are powerfull tools without using any reference that could confirm that the use of frequencyand/or severity is more suitable than quantity.

To clarify, in the manuscript we did not state that these tools were more powerful, rather we stated that there are problems when making comparisons between alcohol, drugs, gambling and smoking in relation to quantity. In behavioural addictions, such as gambling, quantity is not easily comparable to other addictions such as quantity of alcohol consumption, hence the decision to only explore frequency and severity. This is consistent with other reviews of substance and behavioural addictions in which symptom severity and frequency have been the primary outcomes of interest (see example below). These have also been added in the manuscript.

https://www.frontiersin.org/articles/10.3389/fpsyt.2018.00095/full?report=reader#h3

#8 - Along the responses to the questions made, the authors demonstrated that the aim of the review would be demonstrating "the importance of reviews to idenfity gaps in the evidence base, thus calling for the need for researchers to redress these gaps". This is what the authors should have proposed once their results are basically related to it. However, the authors affirm along the text that this review presented confirm the importance of PNF despite the gaps found, and sometimes affirming its efficacy taking in consideration just one study (for gambling). I still sugest to the authors to review their aims for this review, I believe those aims clould be changed to a better result and better discussion.

The aims are as stated in the Prospero registration and therefore cannot be changed. We have re-read the results and discussion and believe that we have interpreted the results of the meta-analysis correctly and not overstated the conclusions.

#9 - I stated that the the authors opted to convert the medians of the studies used into medians, and this choose was not clear. The authors answered that they did it to allow the meta-analysis. Moreover, the authors answered that this is not the standard practice for systematic reviews but acceptable according to their reference, without explain why they chose this option instead of the standard practice for systematic reviews.

As stated in our previous response, the Cochrane Handbook for Systematic Reviews states that this is an acceptable approach. To ensure this process was rigorous we also reported additional sensitivity analyses to assess whether papers for which we converted medians to means affected our findings. Note that the results of these sensitivity analyses were mostly consistent with the main findings, suggesting that this approach did not have a major impact on the results. 

#10 - I noticed that the authors used data derivative from graphics and this was one of the points I highlighted, hoping it was just a misunderstanding. However, the authors affirmed my suspicions answering that in fact they extracted some data from graphical illustrations in rare cases. In my point of view, this is a wrong way to collect data and may compromise the final results.

This was done for one study where no data was provided in the text. As stated in the Cochrane guidelines, extracting data with a ruler from a figure is an acceptable approach. https://training.cochrane.org/handbook/current/chapter-05

#11 - Finally I had highlighted a sentence that was completely wrong and conceptually misunderstood - "Studies of other illicit drugs are notably absent, and researchers continue to be challenged by recruitment and other practical obstacles to adressing behaviours that participants know are against the law". I spent time explained in few lines how wrong this sentence is, but it looks like that the authors do not realize how stigmatized and wrong this sentence is once they changed this sentence for "Studies of other illicit drugs are notably absent, and researchers continue to be challenged by recruitment and other practical obstacles to addressing illicit behaviors". I will not extend myself explaining it again, it was well explained the other time, with no understanding by the authors. My concern is that this sentence was not a wrong way to express the gap in literature relating PNF and illicit drugs, but that this is the authors opinion about people with drug-related problems.

We have changed this to refer to people using other illicit drugs so as to avoid any implied stigma, and have removed reference to this being an illicit behaviour.

Reviewer #3: The systematic review and meta-analysis by Saxton, Rodda and colleagues assessed the efficacy of 'pure' personalized normative feedback (PNF) interventions and in combination with additional self-directed interventions (‘mixed’ PNF) on frequency and symptom severity of hazardous use of alcohol, tobacco, illicit drugs and problem gambling. Authors reported significant effects of 'pure' and ‘mixed’ PNF in reducing short-term frequency and symptom severity associated with alcohol use. In addition, they observed that ‘mixed’ PNF reduced gambling symptom severity in the short term. No effect of the intervention was found on outcomes associated with illicit drugs use. No tobacco study met the inclusion criteria. Authors discussed the importance of conducting future studies to examine whether PNF could be a useful kick start tool to more intensive intervention, or a standalone intervention in some cases. Authors also discussed the need for future cost-effectiveness research to evaluate whether application of PNF in a more comprehensive manner is worthwhile.

I consider that the manuscript is well written, the aims are well defined, methodology is clear and conclusions are supported by data. Authors appropriately discussed the strengths and limitations of the study.

I believe that the previous review brought improvements to the manuscript.

Thank you for your feedback on our manuscript

I would like to point out some minor issues and suggestions:

1. Please, include in the abstract what the acronym “RCT” means (= randomized controlled trial).

The change has been made to the abstract to spell out RCT.

2. In the abstract, authors conclude, “Our findings highlight the efficacy of PNF to address alcohol frequency and symptom severity”, but what about gambling, since they affirmed “PNF with additional interventions reduced short-term gambling symptom severity”. I understand that authors did not draw a firm conclusion about the effect of PNF on gambling symptom severity due to the reduced number of gambling studies included in the analyses, but I think that this could be explained in the abstract.

We have now addressed this in the abstract by highlighting this issue with reduced number of studies.

3. Suggestions: on page 3 (introduction), authors could maybe introduce drugs of abuse topics first (alcohol, tobacco, illicit drugs), and then introduce gambling-related topic.

In addition, on page 10, I think authors could place the sentence “We selected frequency and severity as they are indices that can consistently be applied across all of the included substance and behavioral addictions” after mentioning both evaluated outcomes, frequency and symptom severity (at the end of the paragraph).

We have altered the order of the topics at the start of the introduction as well as the order in the first paragraph. As suggested, we have also relocated the sentence on page 10 to the end of the paragraph.

4. On page 10, authors could include a brief explanation about why exactly they decided to exclude quantity measures, BAC, attitudinal change and dollars spent on gambling.

We have added the following explanation on page 10: ‘In relation to symptom severity we excluded quantity measures, blood alcohol content (BAC) and dollars spent on gambling because they are addiction-specific and not applicable across the range of substance use and behavioural addictions our review included. We also excluded measures of attitudinal change because our aim was to assess the effect of PNF on behavior change. We focused on frequency and severity outcomes as they are indices that can consistently be applied across all of the included substance and behavioral addictions.’

5. I think that table 5 (page 38) may have formatting issues.

Thank you the column had narrowed and separated the minus signs from the text. We have corrected that now.

6. In the “Implications for clinical practice” section (discussion), authors discuss about whether PNF effects in reducing alcohol frequency and symptom severity are meaningful or not, due to the small effect sizes observed. About this, authors brought considerations that were addressed by other reviews. However, it was not clear to me the authors’ opinion on this.

We have reworded the next sentence so as to present evidence to support intervention research that produces moderate effect sizes.

7. In the “Follow up period” section (discussion, page 52), it would be nice to discuss more about the long-term effect of mixed PNF on alcohol symptom severity favoring the control group. Did authors hypothesize why this happens?

Reviewer 3 raises an interesting question. The two long-term alcohol studies were LaBrie et,al 2013 and Johnson et al 2018, and both also report medium-term findings. Forest plots show their findings are already favouring the control group at this earlier stage, in contrast to the other seven studies in the medium-term group. This leads us to conclude their results at 12-23 months were a continuation of the effects seen at 4-11 months. Though there are a number of possible reasons why these two studies demonstrate more favourable outcomes in the control group (e.g., despite randomisation, pre-existing differences between the intervention and control groups affected the results), we feel the finding highlights the caution needed in drawing firm conclusions when relying upon only two studies. We have checked the manuscript to ensure this caution is highlighted for results based on two studies, and have added text to reflect this in the ‘Comparison to wider literature’ section of the discussion.

Reviewer #4: Saxton and colleagues report a systematic review and meta-analysis of the efficacy of Personalized Normative Feedback (PNF) interventions across addictions in meticulous detail based on about 30 studies. This revised manuscript represents a carefully conducted study on an interesting, albeit rather weak intervention to address a variety of addictions. The authors were

responsive to the comments from previous reviewers and made necessary changes, while avoiding recommendations that were inconsistent with their registration of the study in Prospero.

Thank you for your positive comments on our manuscript

The write up of this study was very complete (~65 pages plus supplemental material), but tedious to read. It seemed that many paragraphs were essentially redundant with the substitution of another analysis or sub-analysis with slightly different statistical results. In contrast to Tables 3-6, I found that the forest plot figures in the Supplemental Material (Appendices D and E) really told the whole story. They present the meta-analytic results clearly and much more succinctly than the pages of narrative that were highly redundant with the Tables. Although the figures would need to be redone to save space, the graphic depiction of results is rather compelling, negating the need for the repetitive statistical results format presented.

Thank you to the reviewer on their suggestion to replace the tables with forest plot figures. We agree this is a much improved representation of the findings and have now removed the tables and replaced them with forest plots. To save space, these forest plots were re-done to include only the results of the main analyses. All subgroup analyses have now been reported in text.

The discussion was a rather lengthy recap of results, some of which did not need to be repeated. After focusing on effect sizes, I was anxious to read the authors’ interpretation of the extent to which PNF produced clinically- or societally-meaningful improvements in health and wellbeing. I felt the discussion pertained more to the limitations of the available research, or lack thereof, than the intervention itself. What I took away was that perhaps this may be cost-effective because the small and short-term effects (never clearly defined) are offset by the low cost of the intervention.

Thank you to the reviewer for highlighting the repetition. We have removed the ‘strengths’ section from the limitations and focused this section solely on limitations of the current paper. We also removed 3 paragraphs (400 words) on limitations of the literature, which are discussed elsewhere.

In summary, I believe this review is worthy of publication. This is a carefully done review and meta-analysis and it more than adequately summarizes what is known about the efficacy of Personalized Normative Feedback interventions. However, I am not sure that it warrants such a lengthy treatise.

Thank you to the reviewer for their comments.

---

## [Decision Letter · Decision Letter 2]

13 Jan 2021

PONE-D-20-19468R2

The efficacy of Personalized Normative Feedback interventions across addictions: A systematic review and meta-analysis

PLOS ONE

Dear Dr. Rodda

Thank you for submitting your manuscript to PLOS ONE. After careful consideration, we feel that it has merit but does not fully meet PLOS ONE’s publication criteria as it currently stands. Therefore, we invite you to submit a revised version of the manuscript that addresses the points raised during the review process.

I appreciate all your efforts in reviewing the manuscript. However,  I agree with one of the reviewers' main concerns about the manuscript's length, the redundancy between tables, figures, the narrative, and the unnecessary detail provided. Please, take into consideration these observations in the revised version. 

We look forward to receiving your revised manuscript.

Kind regards,

Fabio Cardoso Cruz, PhD

Academic Editor

PLOS ONE

Reviewers' comments:

Reviewer's Responses to Questions

**Comments to the Author**

1. If the authors have adequately addressed your comments raised in a previous round of review and you feel that this manuscript is now acceptable for publication, you may indicate that here to bypass the “Comments to the Author” section, enter your conflict of interest statement in the “Confidential to Editor” section, and submit your "Accept" recommendation.

Reviewer #1: All comments have been addressed

Reviewer #3: All comments have been addressed

Reviewer #4: (No Response)

2. Is the manuscript technically sound, and do the data support the conclusions?

Reviewer #1: Yes

Reviewer #3: Yes

Reviewer #4: Yes

3. Has the statistical analysis been performed appropriately and rigorously? 

Reviewer #1: Yes

Reviewer #3: Yes

Reviewer #4: Yes

4. Have the authors made all data underlying the findings in their manuscript fully available?

Reviewer #1: Yes

Reviewer #3: Yes

Reviewer #4: Yes

5. Is the manuscript presented in an intelligible fashion and written in standard English?

Reviewer #1: Yes

Reviewer #3: Yes

Reviewer #4: Yes

6. Review Comments to the Author

Reviewer #1: Saxton, Rodda and colleagues re-submitted the paper about systematic review followed by meta-analysis of the efficacy of Personalized Normative Feedback (PNF) alone, and with additional interventions (mixed PNF interventions) for addictive behaviors. The findings are very interesting and the authors presented the PNF as an useful low cost intervention capable to reduce, in some cases, addiction-related behaviors. Regarding the aims proposed by the authors the work was well conducted and data was discussed carefully.Furthermore,authors improved the manuscript including suggestions and clarifying some questions highlighted by the reviewers.

Reviewer #3: (No Response)

Reviewer #4: As I indicated in my previous review, this manuscript reports a systematic review and meta-analysis of the efficacy of Personalized Normative Feedback (PNF) interventions. My main concern was the length of the manuscript, the redundancy between tables, figures, and the narrative, and the unnecessary detail provided, especially when there was so few studies pertaining to subgroup analyses.

This revision did not meet my expectations. For example, the maximum number of studies included in any analysis was evident in Figure 5. There were 0 studies on tobacco, 2 on gambling, 2 on illicit drugs, and 11 on alcohol. In that analysis of symptom severity there were no significant differences between PNF and the control condition. As can be seen in Figures 2-5, the few significant, albeit small effects were predominantly short-term and most evident for alcohol use. Yet the results narrative goes on for 11 pages (not including the 4 forest plot figures and 12 pages for 2 tables). Furthermore, the 10 pages of discussion shares a lot of redundancy with the results. The length of the manuscript is not shorter. Perhaps the authors should have been explicitly told that many of the details and subanalyses should be relegated to supplemental material.

In summary, I believe that this review was carefully conducted. It addresses an important problem and assesses the efficacy of an intervention worthy of investigation. Unfortunately, because so few studies have applied PND to addictions beyond alcohol, it is underpowered and perhaps premature in the other areas examined, especially for tobacco addiction, for which no acceptable studies were found. Because the results are reported in excrutiating detail, it is hard to see the forest for the trees. The authors are a little generous in their conclusions. My interpretation of the results of their review is that although PNF has the advantage of being an inexpensive intervention, it rarely has been evaluated outside of alcohol use among college students; and generally PNF shows little or no effects and those effects are not sustained. If I am missing some additional highlights, they have been buried in a manuscript that still numbers 58 pages (plus figures and supplemental material). This manuscript is acceptable scientifically. The decision about publication rests with the editors who must decide if this modest revision warrants publication in PLOS-One in its present form.

7. PLOS authors have the option to publish the peer review history of their article (what does this mean?). If published, this will include your full peer review and any attached files.

Reviewer #1: No

Reviewer #3: No

Reviewer #4: **Yes: **Howard Goldstein

---

## [Author Response · Author response to Decision Letter 2]

7 Feb 2021

Reviewer #4: As I indicated in my previous review, this manuscript reports a systematic review and meta-analysis of the efficacy of Personalized Normative Feedback (PNF) interventions. My main concern was the length of the manuscript, the redundancy between tables, figures, and the narrative, and the unnecessary detail provided, especially when there was so few studies pertaining to subgroup analyses. This revision did not meet my expectations.

-We are sorry that Reviewer 4’s expectations for the revised version were not met. We have subsequently further reduced the length of the main manuscript by 14 pages – focusing on the results and discussion sections as indicated in the reviewer’s comments. We have shifted text describing the intervention to S3 Appendix C (approx. 3 pages), and the results of sub-group and sensitivity analyses to appendices S3D and S3E (approx. 9 pages). Please see pages 28-42, of the version with track changes highlighted. We also removed approx. 2 pages from the discussion, chiefly from pages 43-49 where results were repeated in the sections comparing our findings to other studies. We also made some minor modifications to the remaining text – all shown in track changes.

For example, the maximum number of studies included in any analysis was evident in Figure 5. There were 0 studies on tobacco, 2 on gambling, 2 on illicit drugs, and 11 on alcohol. In that analysis of symptom severity there were no significant differences between PNF and the control condition. As can be seen in Figures 2-5, the few significant, albeit small effects were predominantly short-term and most evident for alcohol use. Yet the results narrative goes on for 11 pages (not including the 4 forest plot figures and 12 pages for 2 tables). Furthermore, the 10 pages of discussion shares a lot of redundancy with the results. The length of the manuscript is not shorter. Perhaps the authors should have been explicitly told that many of the details and subanalyses should be relegated to supplemental material.

-We feel that reviewer 4’s summary above of our findings reflects what we have communicated in the paper. Our aim was to provide a comprehensive review, regardless of the direction, strength or duration of any intervention effects. However, we also understand from Reviewer 4 and the editor that we could move more of the results section to the appendices to provide a more streamlined main manuscript, to avoid repetition, and improve the reading experience. As stated above, we have moved 12 pages of results to appendices, and have deleted approx. 2 pages of repetitive discussion.

We have retained tables 1 and 2 in the results – which summarise the included studies – because we feel these are important for readers to refer to within the main manuscript, particularly now all descriptive text has been removed.

In summary, I believe that this review was carefully conducted. It addresses an important problem and assesses the efficacy of an intervention worthy of investigation. Unfortunately, because so few studies have applied PND to addictions beyond alcohol, it is underpowered and perhaps premature in the other areas examined, especially for tobacco addiction, for which no acceptable studies were found. 

-We agree with all of Reviewer 4’s considered comments above, and feel that our writing of the paper is reflective of the limitations in the literature. When we embarked upon the review, we did not yet know the extent of gaps in the literature for particular addictive behaviours. We feel it is important to highlight these areas, to invite further research, so the potential of this brief low cost intervention can be better understood.

Because the results are reported in excrutiating detail, it is hard to see the forest for the trees. The authors are a little generous in their conclusions. 

As stated above, we have further reduced the length of the main manuscript. Please see pages 28-49 of the version with tracked changes.

-In terms of being a little generous in our conclusions, we interpret that to mean our conclusions go slightly beyond the evidence we generated. We have reviewed the discussion and conclusion sections and feel that we have been very cautious with our language and interpretation. In the section ‘implications for clinical practice’ we do discuss the potential utility and role for PNF, but couched in cautious language, and with reflection on the meaningfulness of such modest effect sizes. 

My interpretation of the results of their review is that although PNF has the advantage of being an inexpensive intervention, it rarely has been evaluated outside of alcohol use among college students; and generally PNF shows little or no effects and those effects are not sustained. If I am missing some additional highlights, they have been buried in a manuscript that still numbers 58 pages (plus figures and supplemental material). This manuscript is acceptable scientifically. The decision about publication rests with the editors who must decide if this modest revision warrants publication in PLOS-One in its present form.

-Again, we agree with Reviewer 4’s interpretation of the findings. We feel it is important that we deliver the comprehensive review of PNF outlined in our aims and objectives, regardless of the strength, direction and longevity of the effects identified. The number and variety of PNF intervention evaluations identified in our search limited our conclusions, but we do not feel this should be a barrier to publication. We feel our review has highlighted important gaps in the PNF literature, which we hope could attract future research conducted in more varied settings and for a wider range of addictive behaviours so that the broader value of PNF can be better understood.

---

## [Editor Report · Decision Letter 3]

24 Feb 2021

The efficacy of Personalized Normative Feedback interventions across addictions: A systematic review and meta-analysis

PONE-D-20-19468R3

Dear Dr. Rodda

We’re pleased to inform you that your manuscript has been judged scientifically suitable for publication and will be formally accepted for publication once it meets all outstanding technical requirements.

Kind regards,

Fabio Cardoso Cruz, PhD

Academic Editor

PLOS ONE

---

## [Editor Report · Acceptance letter]

16 Mar 2021

PONE-D-20-19468R3 

The efficacy of Personalized Normative Feedback interventions across addictions: A systematic review and meta-analysis 

Dear Dr. Rodda:

I'm pleased to inform you that your manuscript has been deemed suitable for publication in PLOS ONE. Congratulations! Your manuscript is now with our production department. 

Kind regards, 

on behalf of

Dr. Fabio Cardoso Cruz 

Academic Editor

PLOS ONE